# Assessing the biomedical applicability of biogenically synthesized AuNPs using *Salvia splendens* extract

Amr Selim Abu Lila[1], Afrasim Moin[1], Asma Ayyed AL-Shammary[2], Nabeel Ahmad[3], Dinesh Chandra Sharma[4,5], Afza Ahmad[6¤], Syed Mohd Danish Rizvi[1]*, Rohit Kumar Tiwari[7]*

1 Department of Pharmaceutics, College of Pharmacy, University of Hail, Hail, Saudi Arabia, 2 Department of Public Health, College of Public Health and Health Informatics, University of Hail, Hail, Saudi Arabia, 3 Department of Biotechnology, School of Allied Sciences, Dev Bhoomi Uttarakhand University, Dehradun, India, 4 Department of Microbiology, School of Life Sciences, Starex University, Gurugram, Haryana, India, 5 Department of Microbiology, School of Medical and Allied Health Sciences, Sanskriti University, Mathura, Uttar Pradesh, India, 6 Department of Public Health, Dr. Giri Lal Gupta Institute of Public health and Public Affairs, University of Lucknow, Lucknow, Uttar Pradesh, India, 7 Department of Clinical Research, Sharda School of Allied Health Sciences, Sharda University, Greater Noida, Uttar Pradesh, India

¤ Present address: Department of Biochemistry, Babu Banarasi Das College of Dental Sciences, Babu Banarasi Das University BBD City, Lucknow, Uttar Pradesh, India
* tiwarirohitkumar04@gmail.com (RKT); sm.danish@uoh.edu.sa (SMDR)

## Abstract

This study reports the multifunctional potential of gold nanoparticles (AuNPs) biosynthesized by using *Salvia splendens* leaf extract (SSLE). The biosynthesized AuNPs were characterized by UV–Visible spectroscopy, transmission electron microscopy (TEM), and dynamic light scattering, followed by the assessment of their anti-cancer, anti-oxidant, anti-inflammatory and anti-bacterial potentials. The biosynthesized SSLE-AuNPs showed a characteristic absorbance peak at 559 nm that corresponds to the surface plasmon resonance (SPR) band of the AuNPs. The zeta potential of SSLE-AuNPs was estimated to be − 21 ± 1.9 mV, and TEM analysis confirmed the particles to be spherical with an average size of 94.8 ± 5.1 nm. The SSLE-AuNPs exhibited dose-dependent antioxidant activity, with IC50 values of 218.5 ± 4.2 µg/mL (DPPH) and 185.3 ± 3.7 µg/mL (ABTS), compared to ascorbic acid (32.1 ± 1.8 µg/mL and 28.6 ± 1.5 µg/mL, respectively. In addition, SSLE-AuNPs exerted potent antibacterial effect against *Staphylococcus aureus* ($MIC_{50}$ 68 ± 2.1 µg/mL) and *Klebsiella pneumoniae* ($MIC_{50}$ 82 ± 2.3 µg/mL), which was comparable to that of the standard antibacterial agent, tetracycline. Moreover, SSLE-AuNPs induced significant reduction in cellular viability of A549 cells at concentrations of 100, 200 and 400 µg/mL, respectively (p < 0.001). Such cytotoxic potential of SSLE-AuNPs was accompanied by considerable instigation of nuclear fragmentation and condensation, caspase activation, and ROS generation in A549 cells. Furthermore, in vitro studies highlighted the anti-inflammatory potential of SSLE-AuNPs on murine alveolar macrophages

**Data availability statement:** All relevant data are within the manuscript and its Supporting Information files.

**Funding:** This work was supported by Scientific Research Deanship at University of Hail-Saudi Arabia (RG-23 179)]. The funders had no role in study design, data collection and analysis, decision to publish, or preparation of the manuscript.

**Competing interests:** The authors have declared that no competing interests exist.

(J774A.1) via deflating inflammatory mediators such as the proinflammatory cytokines. To sum up, the present findings have substantiated the antioxidant, antibacterial, anticancer and antiphlogistic properties of SSLE-AuNPs, paving the way for subsequent investigations into green synthesized nano-formulations.

---

## 1. Introduction

Historically, plants play a major role in the management of various diseases. Furthermore, plant-based therapeutics are considered a major element in many clinical settings [1,2]. A growing body of evidence has revealed that the plant bioactive constituents exhibit complex molecular mechanisms in order to demonstrate their pharmacological potential. Intriguingly, bioactive compounds from plant extract have recently played an imperative role in the fabrication and biosynthesis of nanoparticles [3,4]. The green synthesized nanoparticles have added advantages over the chemically synthesized nanoparticles, due to their ecofriendly nature, safety and augmented therapeutic potential. These properties of green synthesized nanoparticles prompted the researchers to opt for different plant extracts for the preparation of nanoparticles rather than the usage of harsh or harmful chemicals [5,6].

Indeed, it has now been established that chronic prevalence of oxidative stress serves to be a critical driver for lung carcinogenesis along with tumor aggressiveness [7]. Increased reactive oxygen species (ROS) mediated oxidative stress not only triggers the damage of the genetic material but also activates several oncogenes including KRAS and EGFR [8]. Furthermore, increased ROS levels are also associated with imparting resistance towards apoptosis with concomitantly inducing the activation of pro-survival pathways such as PI3K/AKT and NF-κB [9,10]. At molecular level, inflammation within tumor microenvironment is mediated by macrophages and cytokines. These cytokines predominantly include IL-1β, IL-6 and TNF-α among others which promotes angiogenesis, immune evasion and metastasis [11]. The inflammatory cytokine milieu concomitantly with increased ROS amplifies epithelial mesenchymal transition (EMT) and mutagenesis [12].

Recently, there has been a surge in investigating natural compounds and green synthesized nanoparticles for their plausible therapeutic role against various cancers. Various bioactive compounds namely alkaloids, flavonoids and polyphenols present in different plants have been reported for their potent anticancer effects with significantly lesser side effects as compared to their chemotherapeutic counterparts [13–16]. Subsequently, the emergence of green nanotechnology which utilizes extract of plant, bacterial and/or fungal origin to synthesize nanoparticles has undoubtably elevated bioavailability, drug stability and targeted drug delivery [17]. These approaches have not only resulted in lowering the toxicity but are in line with global efforts for achieving sustained biomedical research [18,19].

In recent times, green synthesized nanoparticles have been substantially explored for their therapeutic efficacy in lung cancer because of their biological and physiochemical characteristics. Gold nanoparticles functionalized with plant extract have

been recently reported to induce ROS and caspase mediated cell death in A549 lung cancer cells [20]. Intriguingly, the anti-inflammatory and antibacterial effects of these nanoparticles are also helpful in tackling certain comorbidities namely bacterial infections along with chronic inflammation, which often aids the progression of lung carcinoma [21]. Furthermore, the antioxidant effects of green fabricated gold nanoparticles further alleviate oxidative stress which in turn is a key factor promoting the survival of cancer cells [22]. Synthesis of gold nanoparticles through *S. splendens* leaf extract is focused on investigating a holistic approach for managing lung cancer through targeting antioxidant, anti-inflammatory, antibacterial and pro-apoptotic pathways synergistically [23].

*Salvia splendens* Linn. (Family: Lamiaceae), commonly known "Red sage" or "Scarlet sage," has been used in the different traditional system of medicines for various ailments since ancient times. *S. splendens* grows throughout in Brazil and many other Asian countries such as India and China [24,25]. *S. splendens* has been identified to contain various bioactive components; among them, anthraquinones, flavonoids, flavon-3-ol derivatives, alkaloid, glycosides, tannin, saponin, terpenoids, reducing sugar, and steroids serve to be the most biologically active components of *S. splendens* [26]. Owing to these bioactive components, *S. splendens* extract has exerted various pharmacological attributes including cardioprotective, hepatoprotective and immunoregulatory, along with the capability to alleviate insulin resistance *in vivo* [27]. In addition, recent preclinical studies have underscored the plausible anticancer, as well as anti-bacterial potential of various *Salvia species* extract, beside its antioxidant and anti-inflammatory properties [28,29]. However, only a few studies have been conducted to decipher the multifunctional potential of *S. splendens* extracts on metallic nanoparticles [30].

Recently, research has focused on the biological activities of *Salvia* medicinal plants used in traditional Chinese medicine (TCM). However, to date a scientific survey of the genus *Salvia* in Chinahas not been carried out. It is now established that *S. splendens* possess a diverse array of terpenoids, flavonoids and phenolic acids including cirsimaritin, salvigenin, apigenin, luteolin, caffeic acid and rosmarinic acid respectively [31]. Intriguingly, it has been reported that the catechol groups in luteolin chelate gold ions not only facilitates the synthesis of nanoparticles but also impart significant antioxidant and pro-apoptotic properties [32]. Salvigenin mediated effects of *Salvia* species is correlated with suppression of IL-6, indicating its anti-inflammatory attributes [28]. Although, *S. officinalis* and *S. miltiorrhiza* synthesized AuNPs have been reported [33], however, the pharmacological attributes of potential *S. splendens* AuNPs remains unexplored despite of its unique phytochemical composition [26].

Metallic nanoparticles such as AuNPs have been previously reported as free radical quenchers [34,35]. Moreover, green synthesized AuNPs by applying various plant-based phytochemicals such as terpenoids and phenolic compounds, were re-ported to augment the antioxidant properties of these nanoparticles [20]. The green synthesis of SSLE mediated AuNPs is achieved by reduction (electron-rich polyphenols such as terpenoids and flavonoids) along with reducing sugar/s present within SSLE that act as electron donor to $Au^{3+}$ ions, facilitating their conversion into $Au^0$ atom [36]. Subsequently, the next step or nucleation is marked by aggregation of $Au^0$ atoms into thermodynamically instable clusters. During the last phase of SSLE mediated AuNPs synthesis, the AuNPs are stabilized from the various proteins and polysaccharides via $-NH_2-COOH$ group interaction which aids the prevention of instable cluster formation [37]. In addition, there are a plethora of reports which have suggested the potent antibacterial activity of plant biosynthesized AuNPs; even against the multi-drug-resistant bacterial pathogens [38,39]. Furthermore, green synthesized AuNPs have also shown strong anti-inflammatory potential, and anticancer activity against different cancer cells. In fact, safety, optical properties, biocompatibility and efficient drug delivery potential of AuNPs have made them the most suitable candidates among different metallic nano-particles for treatment of various ailments.

Indeed, in spite of its diverse pharmacological attributes, *S. splendens*, its ability to synthesize cross-functional AuNPs till date remains under explored. Based on the rich bioactive phytochemical composition of *S. splendens*, this study showcases the green synthesis of SSLE-AuNPs and systematically investigates their therapeutic role. It is hypothesized that SSLE-AuNPs would exhibit enhanced bioactivity owing to its synergy between Au plasmonic resonance and phytochemical based capping. SSLE-AuNPs were comprehensively characterized through DLS and TEM followed by investigation

of their anticancer attributes through ROS generation, caspase-3 activation and modulation of signature inflammatory cytokines. Furthermore, the study also investigates the antioxidant and antibacterial potential of synthesized SSLE-AuNPs for its plausible role as herbal nanotherapeutics.

## 2. Materials and methods

### 2.1 Materials

α,α-Diphenyl-picrylhydrazyl (DPPH), 2,2-Azino-bis (3- ethylbenzthia-zoline-6-sulphonic acid) (ABTS; CAS No. 30931-67-0), gold (III) chloride trihydrate salt, and DAPI dye were procured from Sigma–Aldrich (St. Louis, MO, USA). All the media for microbiology and cell culture, such as nutrient broth medium, Mueller–Hinton agar medium, fetal bovine serum (FBS), DMEM-high glucose medium, antibiotic-antimycotic solution, and ascorbic acid (Vitamin C) were procured from Hi Media, Mumbai, India. The rest of the reagents, solvents, and chemicals used were of standard analytical grade and HPLC grade, respectively.

### 2.2 Methods

**2.2.1. Plant study.** The herbarium of leaves of *S. splendens* were submitted at Indian Institute of Horticultural Research, Hessaraghatta Lake Post, Bangalore-560089 with voucher no: CP/WS/102.

**2.2.2. Strains of bacteria and cell lines.** For antibacterial assessment, strains of *Klebsiella pneumonia* (ATCC 13883) and *Staphylococcus aureus* (ATCC 25923) were used and maintained on nutrient broth media at 37 °C. A549 (human lung cancer) and J774A.1 (murine alveolar macrophages) cell lines were obtained from National Centre for Cell Sciences (NCCS), Pune, India. Both cell lines were cultured in DMEM-high glucose media with 10% fetal bovine serum (FBS) and 1% antibiotic-antimycotic solution under standard conditions.

**2.2.3. Aqueous extraction preparation.** The leaves of *S. splendens* were cleaned and washed with distilled water and dried. The dried leaves (10 g) were crushed in 100 mL of double-distilled water (1:10, w/v) using a mortar and pestle at room temperature. The obtained mixture was filtered through Whatman® filter paper (grade 42) and manually crushed for 15 minutes at room temperature (25±2 °C) to ensure thorough homogenization. The supernatant was collected in a separate tube, filtered and stored at 4 °C until further use. The images of dried leaves, leaf powder and aqueous extract are provided as Fig S1.

**2.2.4. Synthesis and characterization of SSLE-AuNPs.** *2.2.4.1. Green synthesis of SSLE-AuNPs.* 1 mM gold (III) chloride trihydrate salt solution and aqueous extract of *S. splendens* leaves were used to synthesize SSLE-AuNPs by applying the reduction technique, as described previously [40]. The SSLE-AuNPs colloidal suspension was synthesized by reducing 1 mM gold (III) chloride trihydrate (HAuCl$_4$·3H$_2$O, Sigma-Aldrich) with *S. splendens* leaf extract (SSLE) in a 1:1 (v/v) ratio. The reaction mixture (30 mL total volume) was incubated at 40 °C for 24 h until a stable ruby-red dispersion formed, indicating Au$^{3+}$ reduction to Au$^{o}$. The crude suspension was centrifuged (12,000 rpm, 20 min, 4 °C) to pellet AuNPs, followed by discarding the supernatant containing unreacted precursors. The pellet was resuspended in sterile deionized water and subjected to two additional centrifugation cycles to remove residual biomolecules. The obtained filtrate of SSLE-AuNPs was kept at 4 °C until further characterization.

*2.2.4.2. Characterization of SSLE-AuNPs.* Initial confirmation for the reduction of gold chloride salt to SSLE-AuNPs was done through UV-Visible spectrophotometer (UV-1601 PC Series, Shimadzu, Tokyo, Japan) analysis. While, the zeta potential and hydrodynamic radius of SSLE-AuNPs was estimated using Malvern Nano-ZS zetasizer (ZEN3600 Malvern Instrument Ltd., Malvern, UK). Moreover, to analyze the morphology and the actual size of the synthesized SSLE-AuNPs, transmission electron microscope (TecnaiTM G2 Spirit BioTWIN, FEI, Hillsboro, OR, USA) was used. Here, SSLE-AuNPs sample was fixed on carbon coated copper grid prior to TEM analysis.

*2.2.4.3. Fourier transform infrared spectroscopy (FT-IR) analysis.* The FTIR spectra of SSLE and SSLE-AuNPs were analyzed to study the changes in functional groups after formation of AuNPs. Both the extract and synthesized AuNPs were scanned from 4000 to 600 cm$^{-1}$ with a resolution of 4 cm$^{-1}$ using Cary630 FTIR, Agliant Technologies, USA.

 

**2.2.5. Antioxidant assay of SSLE-AuNPs.** *2.2.5.1. DPPH assay.* To investigate the free radical scavenging potential of SSLE-AuNPs, DPPH assay was performed [41]. The percentage (%) antioxidant activity was measured by mixing 1 mL of SSLE-AuNPs (50, 100, 200, 300, and 400 µg/mL) with 100 µM DPPH solution (3 mL). The obtained reaction mixture was vigorously shaken and subsequently incubated for 30 min in the dark. Furthermore, the reduction in optical density as a result of the proton donating ability of SSLE-AuNPs was recorded at 517 nm using a spectrophotometer (BioRad, Hercules, CA, USA). In this assay, an equal volume of methanolic DPPH was taken as control or blank, whereas ascorbic acid was utilized as a positive control. Finally, the radical scavenging ability was computed as follows:

$$Radical\ scavenging\ percentage = \frac{A_0 - A_1}{A_0}\ X\ 100$$

Where $A_0$ is the optical density of blank, $A_1$ is the optical density of DPPH in the presence of different ranges of SSLE-AuNPs or 60 µg/mL ascorbic acid (positive control).

*2.2.5.2. ABTS activity assay.* To measure the antioxidant potential of SSLE-AuNPs, ABTS cation decolorization assay, analogous to DPPH assay, was carried out following the general assay protocol as stated previously [42]. The ABTS•+ radical cation solution was prepared by dissolving ABTS diammonium salt (7 mM) in ultrapure water and oxidized with 2.45 mM $K_2S_2O_8$. The mixture was incubated in the dark for 16 h at 25 °C, then diluted with 50% methanol to an absorbance of 0.70 ± 0.05 at 734 nm. For the assay, 1 mL of SSLE-AuNPs (50–400 µg/mL) was mixed with 3 mL of ABTS•+ solution, incubated for 10 min in the dark, and absorbance was measured at 734 nm. 50% methanolic ABTS solution (non-radical) served as the blank to correct for background absorbance. The scavenging amount of ABTS was assessed using the formula described in the section 2.2.5.1.

**2.2.6. Antibacterial assessment of SSLE-AuNPs.** *2.2.6.1. Qualitative assessment.* The qualitative estimation of antibacterial properties of SSLE-AuNPs was done using agar well diffusion method [43]. The potential of SSLE alone, SSLE-AuNPs and Tetracycline (positive control) was evaluated against *Klebsiella pneumonia* (gram-negative) and *Staphylococcus aureus* (gram-positive) bacterial strains. The dilution (0.5 McFarland) was swabbed onto the surface of Mueller–Hinton agar plates for each strain separately, and four holes of 8 mm diameter were punched to create wells on each inoculated plate. 100 µL of each sample of SSLE, SSLE-AuNPs, Tetracycline, and solvent (negative control) at 50 µg/mL was added to these wells. Plates were incubated at 37 °C for 18 h, and the zone diameter was measured. Experiment was triplicated, and the results were calculated as the mean ± standard deviation.

*2.2.6.2. Quantitative assessment (MIC50).* Quantitative assessment of $MIC_{50}$ value against *K. pneumonia* and *S. aureus* was estimated using broth dilution method [44]. SSLE, SSLE-AuNPs, and Tetracycline (positive control) concentrations were kept between the range of 15.62 µg/mL to 1000 µg/mL, by serial dilution in 96 well microtiter plate. After that 10 µL of tested bacterial strains *K. pneumonia* and *S. aureus* adjusted to 0.5 McFarland standard ($1 \times 10^8$ CFU/mL) was added to each well. The microtiter plates were incubated at 37 °C for 18 h, and $MIC_{50}$ value was calculated for each sample. The experiment was repeated three times to obtain mean ± standard deviation for each value, and the minimal concentration that inhibited the development of each bacterial strain was noted as MIC.

**2.2.7. *In vitro* anticancer activity.** *2.2.7.1. Cytotoxicity assay.* To evaluate the anticancer efficacy of SSLE-AuNPs, MTT assay was performed as per the previously stated protocol [45]. $5 \times 10^3$ A549 and J774A.1 cells/well were cultured with various doses of SSLE-AuNPs (100, 200, and 400 µg/mL) for 24 h. Subsequently 10 µl MTT dye (5 mg/mL) was supplemented in each well and incubated for another 4 h at 37 °C. Ultimately, the purple-colored formazan crystals were solubilized by adding 100 µL DMSO in each well. Then, the 96-well plate was analyzed at 570 nm using a microplate reader (BioRad, Hercules, CA, USA), and the cytotoxicity was computed as percentage (%) cell viability of A549 and J774A.1 cells in comparison to the control cells.

$$Cell\ viability\ (\%) = \frac{Absorbance\ of\ SSLE\ treated\ cells}{Absorbance\ of\ untreated\ cells} \times 100$$

***2.2.7.2. Morphological assessment of SSLE-AuNPs -treated A549 cells.*** A549 cells were cultured with the above-mentioned concentrations of SSLE-AuNPs and were analyzed for their morphological alterations. Briefly, $5 \times 10^3$ A549 cells/well were treated with 100, 200 and 400 µg/mL concentrations of SSLE-AuNPs for 24 h. Finally, changes in the morphology of SSLE-AuNPs -treated A549 cells was visualized under the bright field channel of FLoid imaging station (Thermo-Fischer Scientific, Waltham, MA, USA).

***2.2.7.3. Evaluation of nuclear morphology.*** To qualitatively assess the nuclear condensation and fragmentation in SSLE-AuNPs-treated A549 cells, DAPI assay was performed as stated previously [46]. Briefly, A549 cells were seeded in each well of a 96-well plate, at a density of $5 \times 10^3$, and treated with definite concentrations of SSLE-AuNPs for 24 h. The culture medium was aspirated, and the cells were fixed by adding 100 µL of chilled methanol, subsequently followed by staining with DAPI (2 µg/mL) at 37 °C for 30 min Finally, the treated cells were visualized, and photomicrographs were taken of their fluorescent nuclei using a blue fluorescence channel of FLoid imaging station (Thermo-Fischer Scientific, Waltham, MA, USA).

***2.2.7.4. Caspase-3 estimation.*** The activity of caspase-3 was evaluated, as previously described by using colorimetric caspase-3 specific kit (BioVision, USA) [47]. Briefly, A549 cells were cultured at a density of $3 \times 10^6$ cells and allowed to adhere for 24 h. Thereafter, both SSLE-AuNPs-treated and untreated cells were lysed using ice-cold lysis buffer (50 mL) by briefly incubating them for 10 min on ice. Subsequently, the cell lysate was centrifuged at 10,000 × g for 1 min, and the supernatant was collected and placed on ice. Furthermore, lysate was suspended in 50 mL of cell lysis buffer, after quantifying the protein. 50 µL of reaction buffer constituted by DTT (10 mM) was mixed with 50 µL cell lysate in each well of a 96-well plate. Subsequently, 5 µL DEVD-pNA (4 mM) was added in every well and the plate was incubated additionally for 1 h at 37 °C. Lastly, the plate was read for its absorbance at 405 nm using a spectrophotometer. Percentage (%) change in caspase-3 activity was estimated by collating the results with that of the untreated control.

***2.2.7.5. Effect of caspase-3 inhibitor.*** The cytotoxicity exerted by SSLE-AuNPs against A549 cells was analyzed by using Z-DEVD-FMK (a caspase-3 inhibitor). During the assay, A549 cells were initially cultured with 50 µM Z-DEVD-FMK for 2 h. Furthermore, A549 cells were subsequently treated with different concentrations of SSLE-AuNPs, and again incubated for 24 h. The survivability of A549 cells was measured by using the protocol of MTT assay, as explained in the section 2.2.7.1.

***2.2.7.6. Reactive oxygen species (ROS) assay.*** DCHF-DA dye was used to assess the intracellular ROS levels in SSLE-AuNPs -treated A549 cells both qualitatively and quantitatively [48]. For qualitative assessment of ROS, $2 \times 10^4$ A549 cells were treated with various stated SSLE-AuNPs concentration for 24 h. Subsequently, the cells were stained with DCHF-DA (10 µM) and incubated at for 30 min at 37 °C. The cells were visualized under green fluorescence channel of FLoid imaging station (Thermo-Fischer Scientific, Waltham, MA, USA).

Additionally, the same protocol was repeated during the quantitative evaluation of intracellular ROS levels. The cells were cultured in a 96-well plate (black bottom) and treated with various concentrations of SSLE for 24 h, and then stained with DCFH-DA in dark at 37 °C for 30 min. Lastly, the cells were evaluated for their DCF-DA mediated fluorescence intensity by fluorescent spectrophotometer at an excitation: emission ratio of 485:528 nm. The results were expressed as average fluorescence intensity percentage (%) in comparison to the untreated control A549 cells.

***2.2.7.7. Effects of ROS inhibitor.*** The efficacy of SSLE-AuNPs in instigating ROS production within A549 cells was affirmed in presence of a potent inhibitor of ROS, N-acetyl-L-cysteine (NAC). A549 cells were pretreated with 10 mM NAC for at least 2 h. The cells were then co-cultured with SSLE-AuNPs at the above stated concentration for 12 h. Thereafter, SSLE-AuNPs treated A549 cells were exposed to 10 µM DCFH-DA for 30 min at 37 °C in dark. Finally, the cells were quantified for DCF-DA associated average fluorescence intensity as previously discussed using a fluorescent spectrophotometer.

**2.2.8.** ***In vitro* anti-inflammatory activity of SSLE-AuNPs.** About $1 \times 10^6$ J774A.1 cells/well were cultured in a 6-well plate and were treated with 100 ng/mL LPS for 24 h [49]. The stimulated cells were then cultured with varying doses of SSLE-AuNPs (100, 200 and 400 μg/mL) for additional 12 h in a $CO_2$ incubator. Furthermore, the level of key inflammatory cytokines namely IFN-γ and IL-1β were assessed through ELISA kits (BD Biosciences, San Jose, CA, USA) as per the provided protocol. In addition, LPS stimulated and non-stimulated cells were taken as positive and negative control, respectively.

**2.2.9. Statistical analysis.** The data presented in the present investigation is representative of the mean ± SEM of three individual experiments performed in triplicate through GraphPad Prism (Ver.5). The statistical significance among various treated and untreated sets was estimated by one-way ANOVA and subsequently by Dunnett post hoc test. *$p < 0.05$, **$p < 0.01$ and ***$p < 0.001$ were considered significant.

## 3. Results

### 3.1. Biosynthesis of SSLE-AuNPs

Gold nanoparticles (AuNPs) are generally synthesized by adding a reducing agent to gold (III) chloride trihydrate salt solution, which triggered the reduction of Au ions from $3^+$ oxidation state to 0 oxidation state. This is followed by capping or stabilization of the synthesized AuNPs to prevent their aggregation. Reduction is usually done through chemicals such as sodium borohydrate, hydrazine, and trisodium citrate, which require capping agents such as bovine or human serum albumin to stabilize them [50,51]. However, 'green chemistry' eco-friendly approach for the bio-synthesis of AuNPs through the usage of plant extract or plant enzymes has become a popular choice of the researchers now-a-days to avoid the residual effects of harmful chemicals employed in the chemical mediated synthesis of AuNPs [52–54]. In this study, the enzymes present in *S. splendens* leaf aqueous extract (SSLE) plausibly induced the reduction of gold salt into AuNPs, which was further capped by phytoconstituents present in the extract.

### 3.2. Characterization of biosynthesized SSLE-AuNPs

The reduction process and formation of SSLE-AuNPs was confirmed visibly by a color change from yellow to ruby red. Characteristic Surface plasmon resonance peak of AuNPs during UV-Vis spectrophotometer analysis was observed at 559 nm (Fig 1a). It is a well-established fact that a peak between 500–600 nm characterizes the surface plasma resonance of AuNPs [55,56], which confirmed the successful formation of SSLE-AuNPs.

Dynamic light scattering (DLS) approach is one of the most commonly used technique to estimate nanoparticle size and distribution in an aqueous medium [57,58]. The particle size of the biosynthesized AuNPs estimated by DLS approach was 114.7 ± 9.8 nm with a PDI of 0.30 (Fig 1b), indicating homogeneity of (size) distribution. Zeta potential analysis was also conducted to evaluate the stability of SSLE-AuNPs. Generally, a zeta potential value of ± 20 mV is required to maintain the colloidal stability of the NPs [59,60]. In this study, the zeta potential of the biosynthesized SSLE-AuNPs was found to be –21 ± 1.9 mV (Fig 1c), confirming relative colloidal stability of the synthesized nanoparticles. In addition, the SSLE-AuNPs does not show any aggregation or color change even after keeping them at room temperature for 6 months.

However, the size of inorganic core of SSLE-AuNPs was determined as 94.8 ± 5.1 nm (Fig 1d) by TEM analysis. The average size of SSLE-AuNPs as shown in the size distribution graph, was 90.78 nm (Fig 1e). The smaller size of SSLE-AuNPs, determined by TEM analysis, compared to that estimated by DLS approach might be ascribed to the different approaches used for size measurement, by both the techniques. TEM analysis measures the size of the inorganic core only, whereas DLS includes the adhered solvent layer as well during size measurement. Similar variations in nanoparticle size by different techniques were observed in many recent reports [61–63].

FTIR spectra could be applied to detect the phytochemicals liable for capping or stabilization of metal nanoparticles. Herein, FTIR spectra helps us to assure the attachment of phytochemicals present in SSLE on the surface of AuNPs. The comparative analysis of FTIR of SSLE and SSLE-AuNPs also helped us to understand the changes that occurred

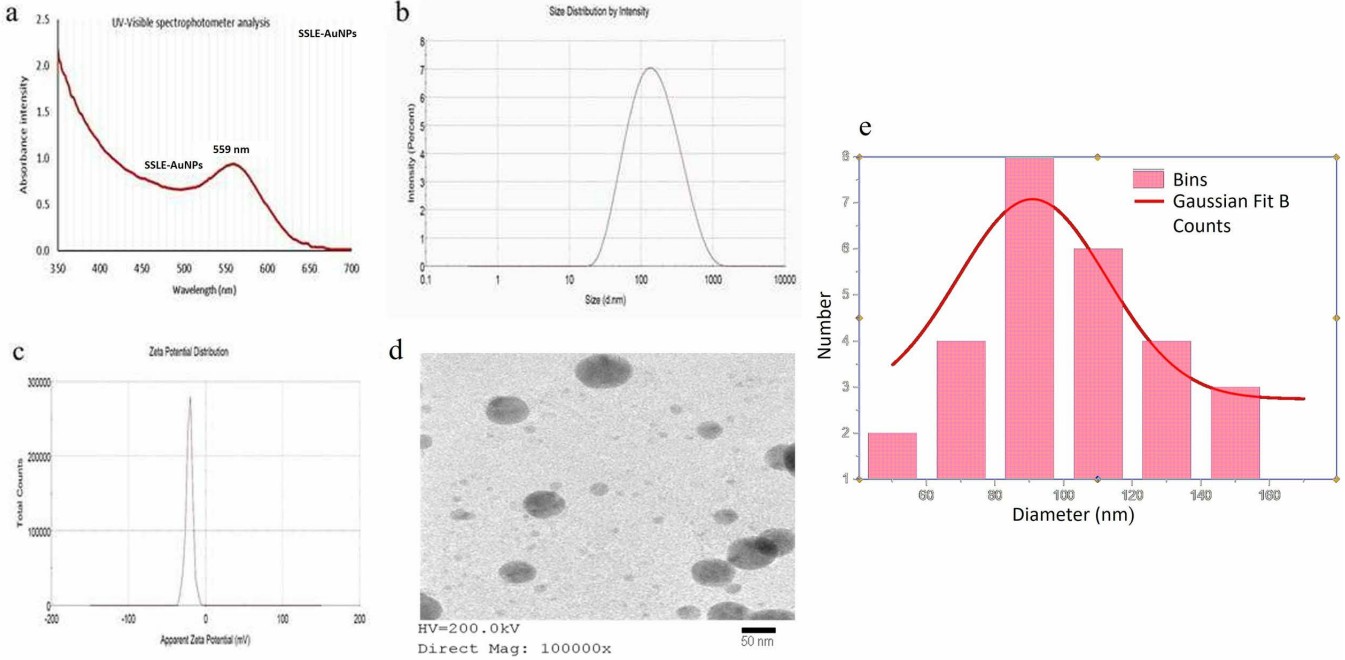

**Fig 1. Characterization of SSLE-AuNPs; (a) UV-Vis spectrophotometry; (b) particle size; (c) Zeta potential; and (d) Transmission Electron Microscopy image (e) Size distribution graph showing the size of SSLE-AuNPs.**

after attachment of SSLE phytochemicals on AuNPs. As shown in Fig 2, the major broad peak in the spectra of SSLE was seen between 3200−3400 cm⁻¹ that corresponds to OH stretching, the peaks between 2700−2950 cm⁻¹ were because of asymmetric and symmetric vibrations of C-H stretching of alkanes, the region between 1600−1700 cm⁻¹ corresponds to the bending mode of N-H and aromatic compounds, and the peaks situated at 1150 cm⁻¹ and 1020 cm⁻¹ are attributed to the C=C bond and the C–OH bonds. On the other hand, SSLE-AuNPs showed a slight shift in OH stretching with enhanced broadening, strong absorption was observed at N-H stretching corresponding to aromatic compounds, reduced peak intensities were seen at C=C and C–OH bonding positions. These slight changes in FTIR pattern are obvious after attachment of phytoconstituents of SSLE onto the surface of AuNPs and specifically, NH moiety role in attachment of bio-active compounds to AuNPs is well known. Thus, it could be inferred that SSLE phytoconstituents have been successfully attached and stabilized the AuNPs.

### 3.3. SSLE-AuNPs exhibited scavenging efficacy of DPPH radicals and ABTS radicals

DPPH is an organic free radical that is highly stable with an absorption width of 512–528 nm. It works on the rationale of deflating the potential of the alcoholic solution of DPPH in the presence of hydrogen donating stimulant. The SSLE-AuNPs exhibited dose-dependent antioxidant activity, with IC50 values of 218.5±4.2 µg/mL (DPPH) and 185.3±3.7 µg/mL (ABTS), compared to ascorbic acid (32.1±1.8 µg/mL and 28.6±1.5 µg/mL, respectively. To explore the free radical scavenging potential of SSLE-AuNPs, DPPH assay was performed. It was evident that SSLE-AuNPs exerted a dose-dependent free radical scavenging potential. SSLE-AuNPs substantially neutralized the DPPH free radicals by 22.36±1.04% (50 µg/mL), 39.41±1.43% (100 µg/mL), 62.40±2.81% (200 µg/mL), 79.56±2.73% (300 µg/mL) and 88.46±2.34% (400 µg/mL), respectively (Table 1).

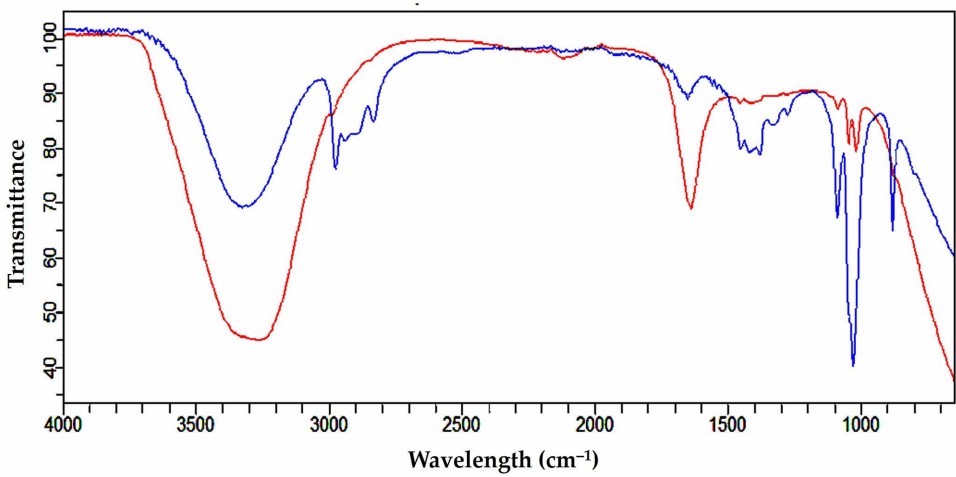

**Fig 2. FTIR spectra of SSLE (Blue line) and SSLE-AuNPs (Red line).**

Similarly, the antioxidant activity of SSLE-AuNPs was further evaluated by ABTS assay. This radical is known for showing absorption in the visible range of 734 nm with a short reaction time. Table 1 demonstrated that SSLE-AuNPs was able to substantially scavenge ABTS radical in a dose-dependent trend by $17.66 \pm 1.34\%$ (50 µg/mL), $49.82 \pm 1.42$ (100 µg/mL), $58.49 \pm 2.32\%$ (200 µg/mL), $70.21 \pm 2.43\%$ (300 µg/mL), and $84.51 \pm 2.59\%$ (400 µg/mL), respectively. Collectively, these results indicated that SSLE-AuNPs hold a promising potential for scavenging ABTS radical.

### 3.4. Antibacterial activity of SSLE-AuNPs

Qualitative assessment through agar well diffusion method verified the ability of SSLE-AuNPs to diffuse through the agar and inhibit the growth of the tested strains. Herein, SSLE alone and tetracycline were applied along with SSLE-AuNPs to provide comparative assessment. It was observed that all the tested samples were more effective against *S. aureus* than *K. pneumoniae* (Fig 3a–3c). SSLE-AuNPs were almost twice more effective than the SSLE alone, however, among the tested samples tetracycline (positive control) showed the maximum potency against both the tested strains. Nevertheless, the difference between the zone of inhibition of SSLE-AuNPs and the positive control was comparable.

Moreover, the quantitative (MIC) analysis was in line with the preliminary qualitative analysis. The $MIC_{50}$ values of SSLE-AuNPs against *S. aureus* and *K. pneumoniae* were estimated as $68 \pm 2.1$ µg/mL and $82 \pm 2.3$ µg/mL, respectively (Fig 4a–4c). On the other hand, $MIC_{50}$ values of tetracycline were $51 \pm 1.4$ µg/mL and $59 \pm 1.9$ µg/mL against *S. aureus* and *K. pneumoniae*, respectively. SSLE-AuNPs showed reduction in the $MIC_{50}$ values by approximately 4–6 times as compared to SSLE alone. Thus, confirming the strong antibacterial potency of the biosynthesized SSLE-AuNPs, which was also comparable with the positive control tetracycline.

### 3.5. Cytotoxic effects of SSLE-AuNPs against A549 lung cancer cells

MTT-based cytotoxicity study was performed to evaluate the growth suppressive effects of SSLE-AuNPs against A549 cell lines. The A549 cell were cultured with varying concentrations of SSLE-AuNPs for 24 h. As sown in Fig 5a, the A549 cell viability was drastically reduced to $79.49 \pm 3.56\%$ (100 µg/mL), $47.82 \pm 5.17\%$ (200 µg/mL) and $24.48 \pm 2.52\%$ (400 µg/mL) in comparison to the untreated A549 cells. We have also compared the cell viability of SSLE-AuNPs with SSLE sample (without AuNPs). It was demonstrated that SSLE alone showed moderate cytotoxicity ($93.15 \pm 2.63\%$, $77.48 \pm 3.43\%$, and $48.48 \pm 2.25\%$ viability at 100, 200, and 400 µg/mL, respectively as shown in Fig 5b) whereas SSLE-AuNPs exhibited

**Table 1. DPPH and ABTS inhibition potential of SSLE-AuNPs and ascorbic acid.**

| SSLE-AuNPs (µg/ml) | DPPH scavenging (%) | ABTS radical scavenging (%) |
|---|---|---|
| 50 | 22.36 ± 1.04* | 17.66 ± 1.34* |
| 100 | 39.41 ± 1.43* | 49.82 ± 1.42* |
| 200 | 62.40 ± 2.81* | 58.49 ± 2.32* |
| 300 | 79.56 ± 2.73** | 70.21 ± 2.43** |
| 400 | 88.46 ± 2.34** | 84.51 ± 2.59** |
| Ascorbic acid (60 µg/mL) | 94.22 ± 1.91** | 96.42 ± 2.12** |

Data reported here represents mean ± SEM (n = 3). *$p < 0.05$; **$p < 0.01$ comparatively with the control DPPH: 2,2-diphenyl-1-picrylhydrazyl; ABTS: 2,2′-Azino-bis (3-ethylbenzothiazoline-6-sulfonic acid) diammonium salt; SSLE-AuNPs: Gold nanoparticles synthesized by *S. splendens* leave extract.

significantly enhanced efficacy at equivalent concentrations. Thus, SSLE-AuNPs demonstrated a significant cytotoxic potential against the proliferation of lung cancer cells. Furthermore, we observed insignificant cytotoxic effects of SSLE-AuNPs in J774A.1 normal murine alveolar macrophages (Fig 5c).

Additionally, phase contrast microscopy was performed to study the substantial changes in the morphology of A549 cells after treatment with SSLE-AuNPs. It was noticed that treatment with SSLE-AuNPs induced swelling, lysis and withering of cell organelles within A549 cells at the all the tested concentrations. In contrast, no significant morphological alteration was noticed in control cells (Fig 6).

### 3.6. Altered nuclear morphology noticeable in SSLE-AuNPs treated A549 cells

Condensation and fragmentation of cellular nuclei are the crucial attributes of apoptosis. In order to analyze the alterations of A549 nuclei after treatment with various concentrations of SSLE-AuNPs, DAPI assay was performed. The captured fluorescent micrographs revealed that SSLE-AuNPs was able to instigate a dose-dependent condensation and fragmentation A549 nuclei in comparison to the untreated cells (Fig 7).

### 3.7. Elevated ROS generation by SSLE-AuNPs

DCFH-DA staining was done to access the consequence of SSLE-AuNPs on ROS generation. The captured fluorescent micrographs deciphered that treatment with SSLE-AuNPs (100, 200, and 400 µg/mL) culminated in enhanced DCF-DA-mediated fluorescence intensity in A549 cell lines. Thus, SSLE-AuNPs were able to substantially elevate the generation of ROS in the studied cells (Fig 8a). Furthermore, the production of intracellular ROS in SSLE-AuNPs -treated A549 cells was quantified to validate our qualitative findings. The intracellular ROS level increased by 38.71 ± 3.86% at the indicated dose of 100 µg/mL, when compared to the untreated cells. However, the amount of ROS subsequently elevated to 74.68 ± 4.65% (200 µg/mL) and 96.42 ± 2.97% (400 µg/mL) in SSLE-AuNPs treated A549 cells (Fig 8b). Thus, SSLE-AuNPs enhanced the amount of intracellular ROS in a dose-dependent trend in the lung cancer cells.

Additionally, to confirm the generation of intracellular ROS in A549 cells, due to treatment with SSLE-AuNPs, the quantitative measurement of ROS was done in presence of NAC (ROS scavenger). Our findings substantiated that pre-treatment with NAC (5 mM) totally diminished the elevated ROS generated within the lung cancer cells in response to SSLE-AuNPs treatment (Fig 8c).

### 3.8. Enhanced caspase-3 activity by SSLE-AuNPs treatment

To investigate the potential of SSLE-AuNPs on the caspase-3 activity in A549 cells, colorimetric based evaluation of caspase-3 was performed. It was noticed that the increasing doses of SSLE-AuNPs was able to elevate the action of

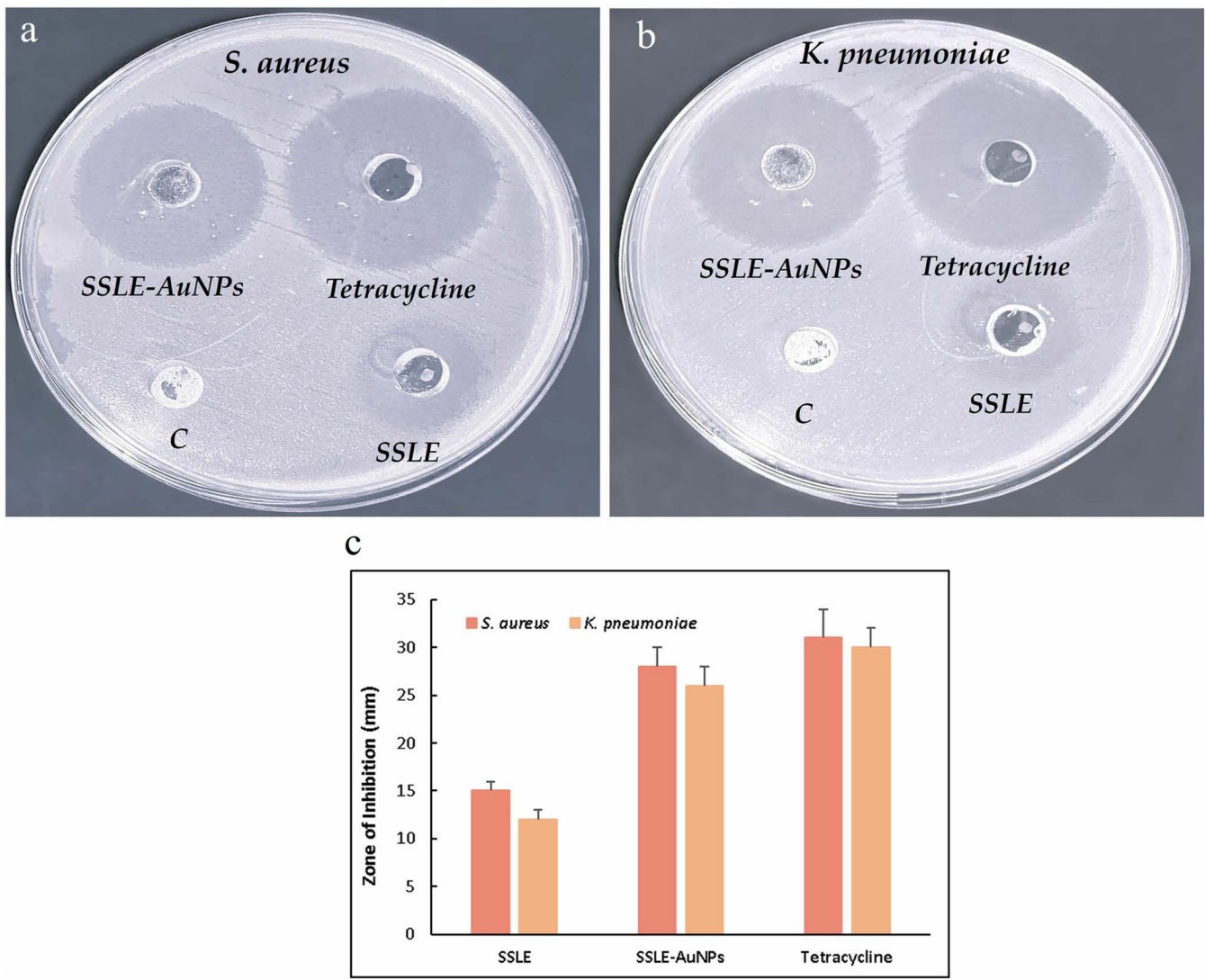

**Fig 3. Qualitative antibacterial assessment of SSLE-AuNPs against (a)** *S. aureus* **and (b)** *K. pneumoniae* **(c) Graph depicting zone of inhibition of SSLE, SSLE-AuNPs and tetracycline.**

caspase-3 in a dose-dependent manner. As evidenced in Fig 9a, the caspase-3 activity in A549 cells was substantially enhanced by 45.36±3.76% (100 μg/mL), 76.79±4.91% (200 μg/mL) and 106.25±4.75% (400 μg/mL), respectively, as compared to the control cells.

In addition, in order to validate that caspase-3 was activated as a result of treatment with SSLE-AuNPs in A549 cells, cell viability assay was performed in the presence of caspase-3 inhibitor. It was noticed that pre-treatment with caspase-3 inhibitor caused a significant decrease in the cytotoxicity of SSLE-AuNPs in A549 cells (Fig 9b). These results suggest that caspase activation plays a crucial role in SSLE-AuNPs-mediated apoptosis.

### 3.9. Amelioration of LPS-mediated inflammatory cytokines by SSLE-AuNPs

Signature proinflammatory cytokines; IFN-γ and IL-1β, were found to be significantly elevated after stimulation with LPS (well-known instigator of inflammatory response) (Fig 10a and 10b). Pretreatment of J774A.1 cells with LPS increased

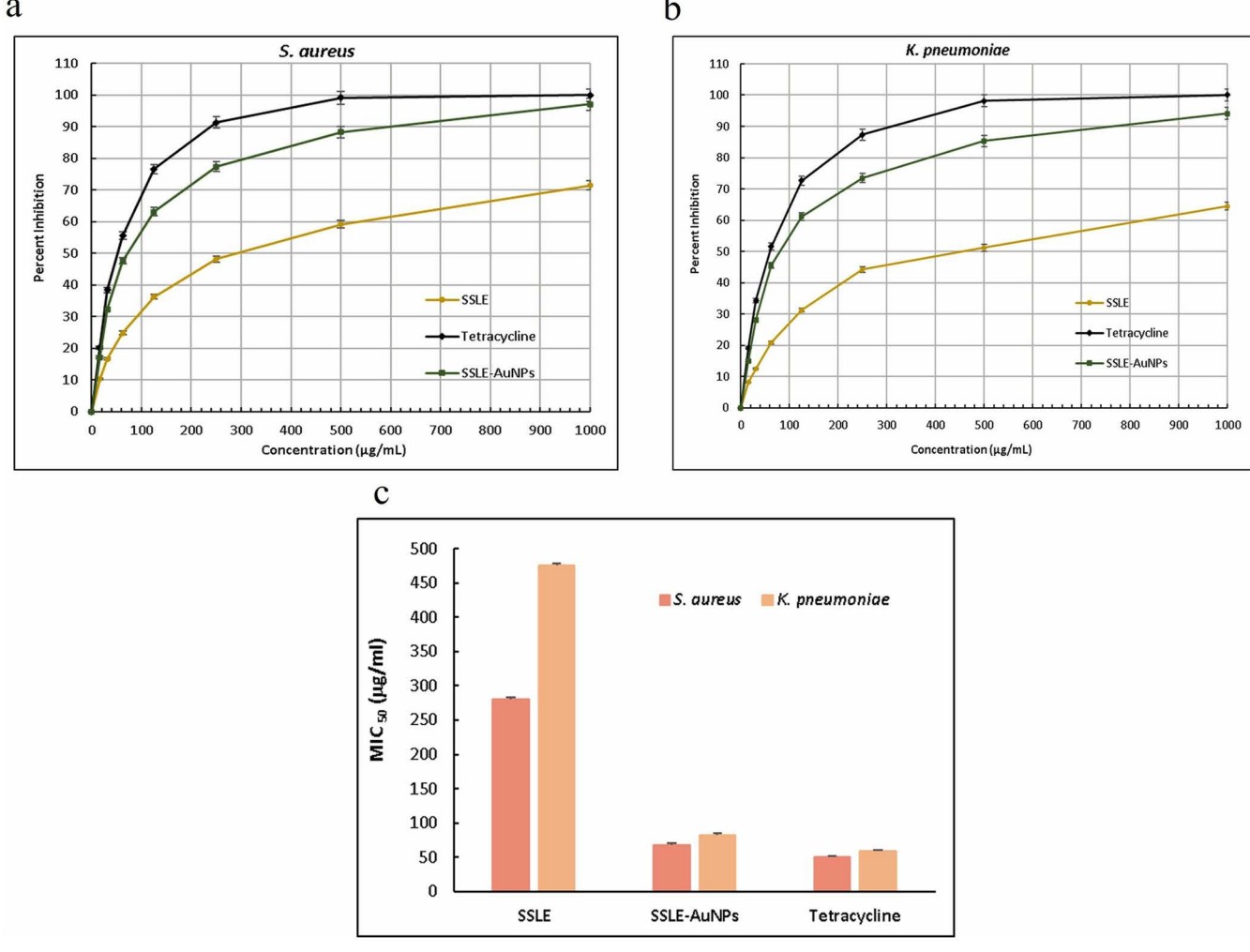

**Fig 4. Quantitative antibacterial assessment (MIC) of SSLE-AuNPs against (a)** *S. aureus* **and (b)** *K. pneumoniae* **(c) Graph representing MIC$_{50}$ values of SSLE, SSLE-AuNPs and tetracycline.**

the levels of IFN-γ and IL-1β by 76.62±5.30 and 73.29±4.30 pg/mL, respectively. In comparison with LPS stimulated J774A.1 cells, SSLE-AuNPs treated cells showed a significantly decline in IFN-γ levels to 49.70±4.54 pg/mL (100 µg/mL), 34.31±4.57 pg/mL (200 µg/mL) and 26.63±2.44 pg/mL (400 µg/mL). Similarly, IL-1β level was reduced to 51.70±4.61 pg/mL (100 µg/mL), 39.98±4.05 pg/mL (200 µg/mL) and 33.96±2.88 pg/mL (400 µg/mL).

## 4. Discussion

The present study explored the multifunctional potential of biosynthesized SSLE-AuNPs by estimating their free radical scavenging, anti-bacterial, anti-cancer and anti-inflammatory activities. Free radicals are known to exert a deleterious effect on several biomolecules such as proteins, lipids, and genetic materials, and subsequently alter the functioning of immune system [64]. Antioxidants serve to be an important means of negating these harmful effects of free radicals, since they not only impede the generation of free radicals, but further inactivate them within the cells. Although, human cells tend to protect themselves against the deleterious effects of free radicals by employing some natural antioxidant

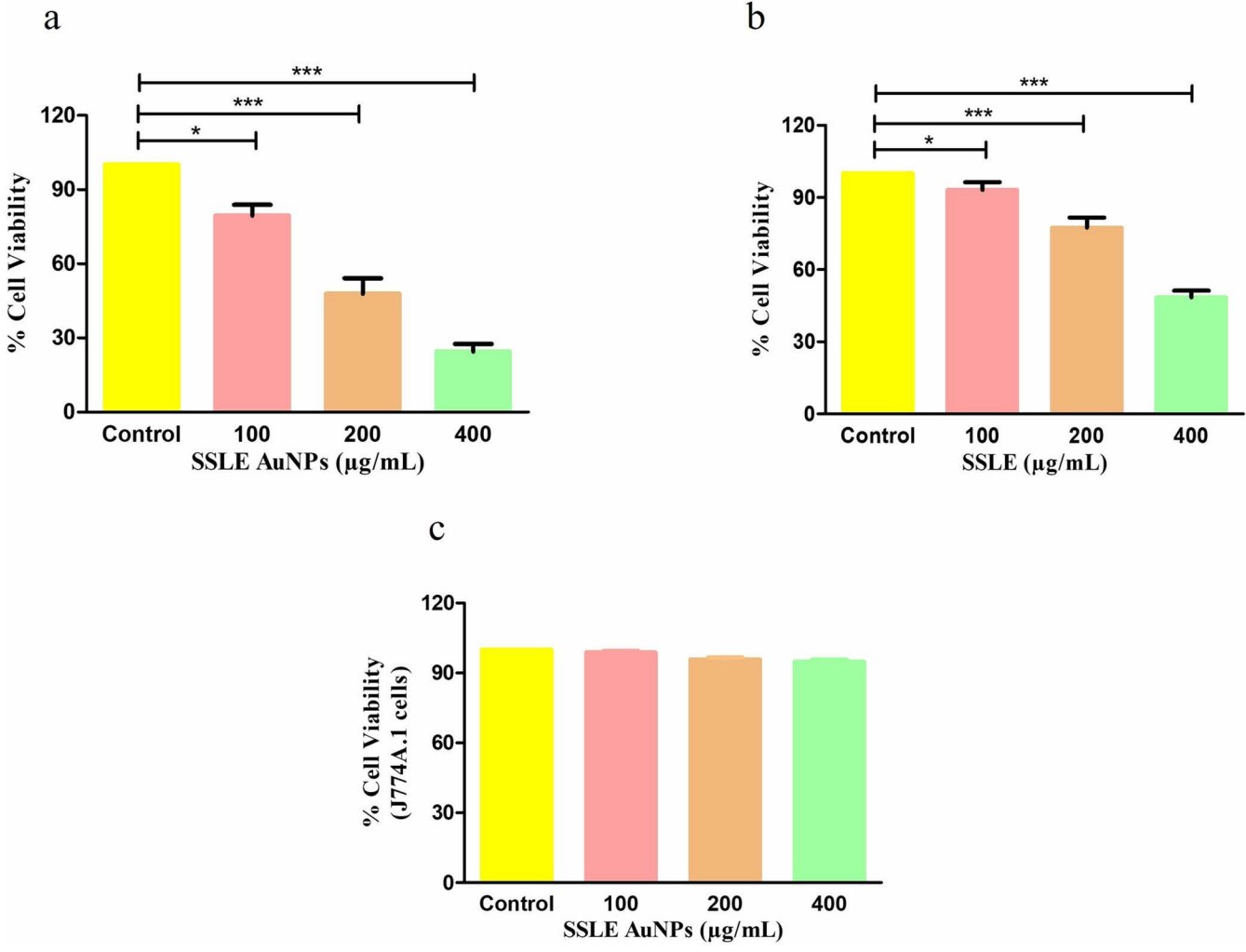

**Fig 5. The cytotoxic effects of (a) SSLE-AuNPs and (b) SSLE on non-small cell lung carcinoma A549 cells as assessed through MTT assay and (c) insignificant cytotoxic effects of SSLE-AuNPs on J774A.1 cells.** *$p < 0.05$, and ***$p < 0.001$.

defense mechanism, however, these occasionally do not function at optimum levels specially during diseased conditions. Thus, external antioxidants are required to substitute for their deficiency under the disease state. Recently, the green biosynthesized AuNPs have emerged as potent free radical scavengers [65–67]. Our study showed that the synthesized SSLE-AuNPs were less than 100 nm in size and were found to be relatively stable as indicated by their zeta-potential. The results obtained during the study also indicated that the synthesized SSLE-AuNPs had potent dose-dependent free radical scavenging potential by inhibiting 88% DPPH at 400 μg/mL concentration. During the study it was also found that SSLE-AuNPs exerted antibacterial effects against both *S. aureus* and *K. pneumoniae* strains. SSLE-AuNPs was further effective in exerting selective cytotoxicity against A549 cells by inducting ROS-mediated apoptotic cell death followed by activation of caspase-3. Furthermore, SSLE-AuNPs also showed their competence in suppressing the levels of signature proinflammatory cytokines namely IFN-γ and IL-1β illustrating their therapeutic versatility.

The antibacterial assessment of SSLE-AuNPs was performed on two strains, i.e., *K. pneumoniae* (gram-negative) and *S. aureus* (gram-positive). Interestingly, both these strains have close association with lung cancer. In fact, several

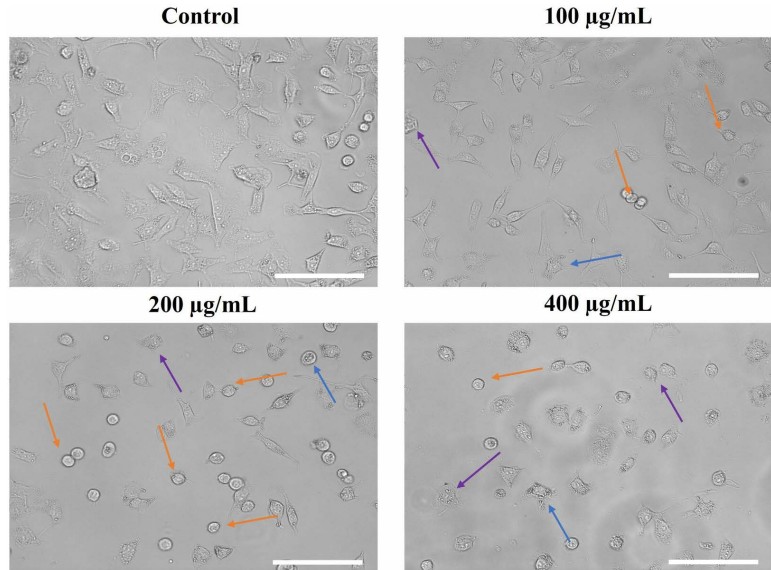

**Fig 6. Morphological alterations in SSLE-AuNPs treated A549 cells.** Orange, blue and purple arrows indicate rounding, blebbing of cytoplasm and rupturing of A549 cells respectively. Scale bar = 100 μm.

re-ports have suggested that *S. aureus* infection is a common complication of lung cancer patient, which can further stimulate the lung cancer proliferation and metastasis [68–70]. Similarly, *K. pneumoniae* also has an association with lung cancer patients [71]. As our study includes lung cancer A549 cells to evaluate the anticancer effects, the anti-bacterial assessment against these two tested strains has its due relevance. The well diffusion (Fig 3a and 3b) and MIC$_{50}$ (Fig 4) clearly indicated that SSLE-AuNPs were several times more potent than SSLE alone (extract alone), further, their activity was comparable with the positive control, tetracycline. It is evident from recent mechanistic reports on AuNPs that AuNPs could be a potent tool to fight against resistant bacterial infections [72]. Intriguingly, SSLE-AuNPs showed superior antibacterial activity (MIC 68 μg/mL vs. *S. aureus*) compared to AuNPs from *S. officinalis* (MIC 120 μg/mL) likely due to differences in phytochemical composition [73,74]. In addition, it is observed in the present investigation that all the tested samples (SSLE, SSLE-AuNPs and Tetracycline) were more effective against gram-positive *S. aureus* than gram-negative *K. pneumoniae*. This might be due to structural differences between gram-positive and gram-negative bacterial strains; gram-negative is usually regarded as more resistant than gram-positive bacterial strains due to the presence of extra outer membrane [74].

After obtaining promising antioxidant and antibacterial results, the quest for multipotent ability of SSLE-AuNPs prompted us to explore it for anticancer potential. There is a plethora of articles that showed potent anti-lung cancer activity of extracts obtained from *S. splendens* [75,76]. However, none have used its aqueous extract to biosynthesize AuNPs, and tried to develop them into potent anticancer agent against lung cancer. Herein, SSLE-AuNPs were developed, characterized, and tested against A549 lung cancer cells to evaluate their anticancer potential. In the present report, SSLE-AuNPs were shown to exerts cytotoxic effects against human non-small cell lung carcinoma (A549) cells (Fig 5a). Morphological studies also showed that post-SSLE-AuNPs exposure, shrinkage of cells increased with concomitant elevation in blebbing of plasma membrane (Fig 6). Furthermore, SSLE-AuNPs also induced fragmentation and condensation of nuclear content with A549 cells (Fig 5) indicating onset of apoptotic cell death [77]. As discussed above, oxidative stress results from disparity between the synthesis and removal of free radicals [78]. Reactive oxygen species (ROS) is a common by-product of oxidative stress, and are chemically small, unstable molecules constituted by singlet oxygen, hydroxyl

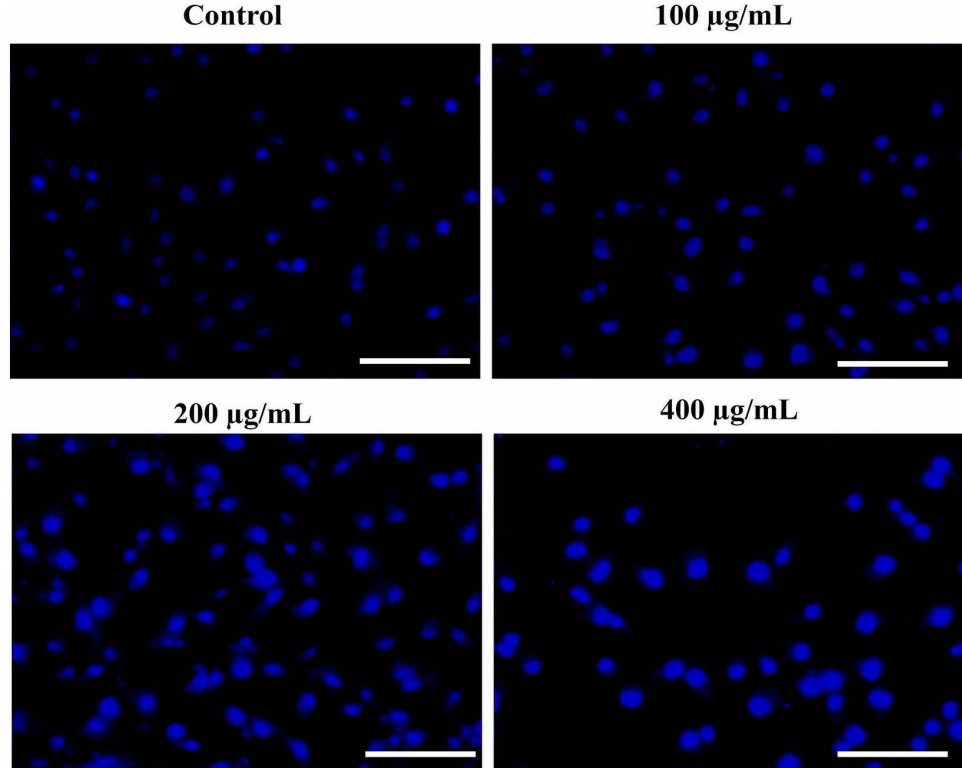

**Fig 7. DAPI assay for analysing SSLE-AuNPs induced nuclear condensation and fragmentation in human lung cancer A549 cells post-SSLE exposure.** Scale bar = 100 µm.

and superoxide anion radical. Several reports have shown that acute elevation in intracellular ROS, either directly or indirectly damages the lipids, proteins, and nuclear content of a cell, resulting in the activation of apoptotic cell death [79]. DCFH-DA staining was used to study the ROS generation in A549 cells. Here, intrinsic ROS was increased significantly post treatment with SSLE-AuNPs (Fig 8b). Moreover, preliminary treatment with NAC attenuated the ROS generation in SSLE-AuNPs-treated A549 lung cancer cells, which subsequently confirmed our findings that SSLE-AuNPs induced ROS generation within A549 cells (Fig 8c). Thus, on the basis of the results obtained, it could be proposed that SSLE-AuNPs have the dual ability, i.e., 1) to quench free radicals like antioxidants under normal circumstances; 2) to augment the levels of ROS in cancer cells. Further-more, caspases play critical role in apoptosis and belong to the class of cysteine prote-ases. Our findings have revealed that SSLE-AuNPs has increased the activity of caspase-3 in A549 lung cancer cells (Fig 9). In fact, caspase-3 and ROS-based apoptosis activation is a well-reported strategy followed by green synthesized AuNPs to target cancer cells [80,81]. Thus, it can be safely stated that SSLE-AuNPs successfully induced apoptosis in lung cancer cells via generation of ROS and activation of caspase 3.

Finally, SSLE-AuNPs were evaluated for anti-inflammatory potential as well. Acute inflammatory response serves as an impetus for resolving the pathological infections within the human body. On the contrary, chronic inflammatory response lays the foundation of several autoimmune disorders including arthritis, Alzheimer's and Parkinsonian disease [82]. A major cause underlying chronic inflammation is the accumulation and/or hyper-activation of proinflammatory cytokines namely IL-1β and interferon (IFN-γ) among several others resulting in modulation of NF-κB and MAPK pathways [82,83]. The present study examined the anti-inflammatory potential of SSLE-AuNPs in vitro using murine alveolar macrophages (J774A.1). J774A.1 cells pre-stimulated with LPS showed significant increase in quantity of inflammatory cytokine markers namely IL-1β and

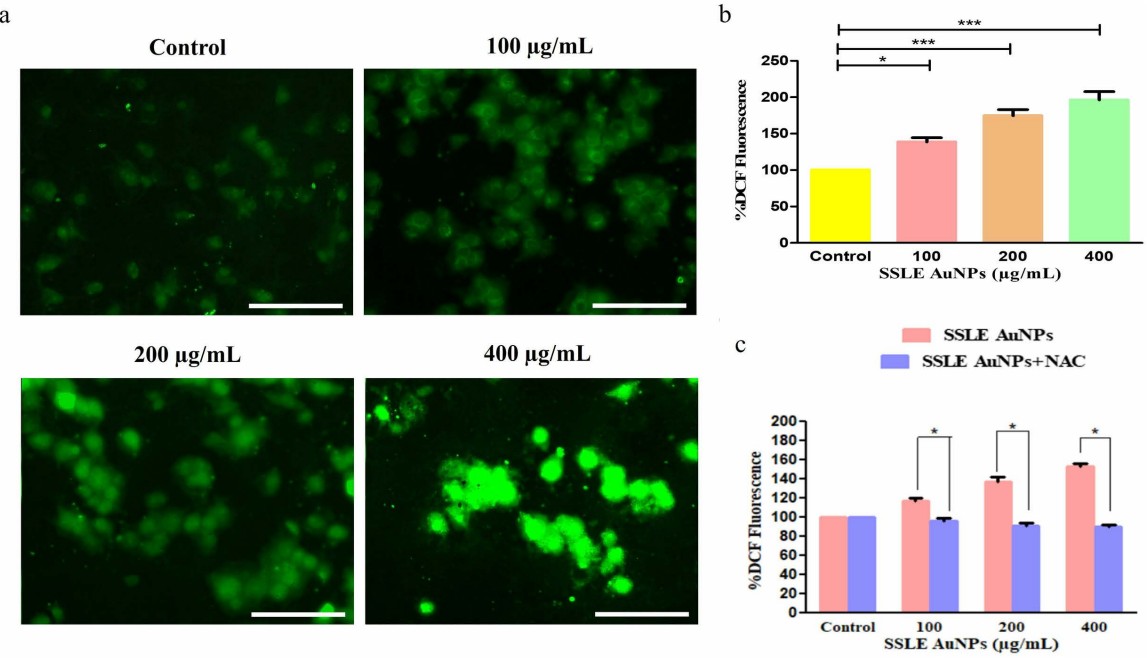

**Fig 8. SSLE-AuNPs induced effects on generation of intracellular ROS levels (a) assessed qualitatively; (b) quantitatively through DCF-DA staining and (c) effect of NAC in ameliorating SSLE-AuNPs induced ROS generation in human lung cancer A549 cells.** Scale bar = 100 μm, *$p < 0.05$, and ***$p < 0.001$.

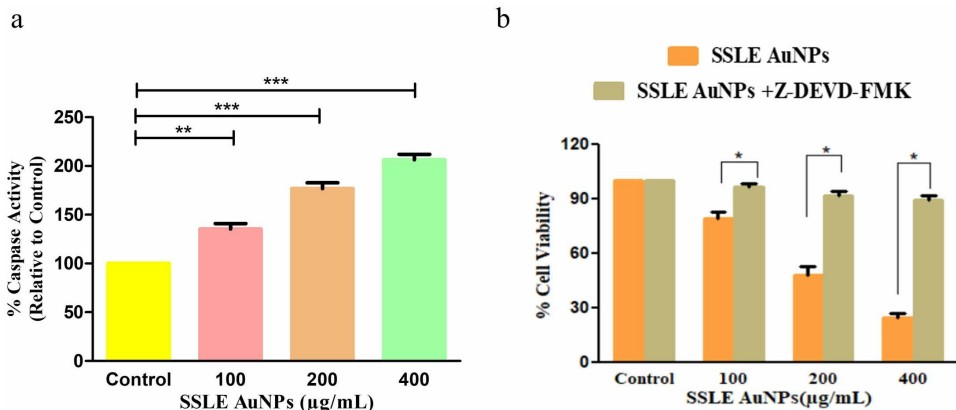

**Fig 9. (a) Activity level of caspase-3 and (b) effect of caspase-3 inhibitor on the cellular viability of human lung cancer A549 cells post-SSLE-AuNPs exposure.** *$p < 0.05$, **$p < 0.01$ and ***$p < 0.001$.

TNF-α. However, SSLE-AuNPs exposure led to a substantial decline in the amount of these inflammatory cytokines (Fig 10), indicating their anti-inflammatory potential. Recently, similar observations of downregulation of IL-1β and TNF-α after treatment with AuNPs have been reported that support the present study findings. Based on the results obtained, it is proposed that SSLE-AuNPs could be further exploited to be used as a multi-potent therapeutic candidate, specifically for the treatment of lung cancer in near future. However, without *in vivo* data and toxicity analysis, the applicability of SSLE-AuNPs is still questionable. Nevertheless, the promising data of the present study might pave the way for future directions.

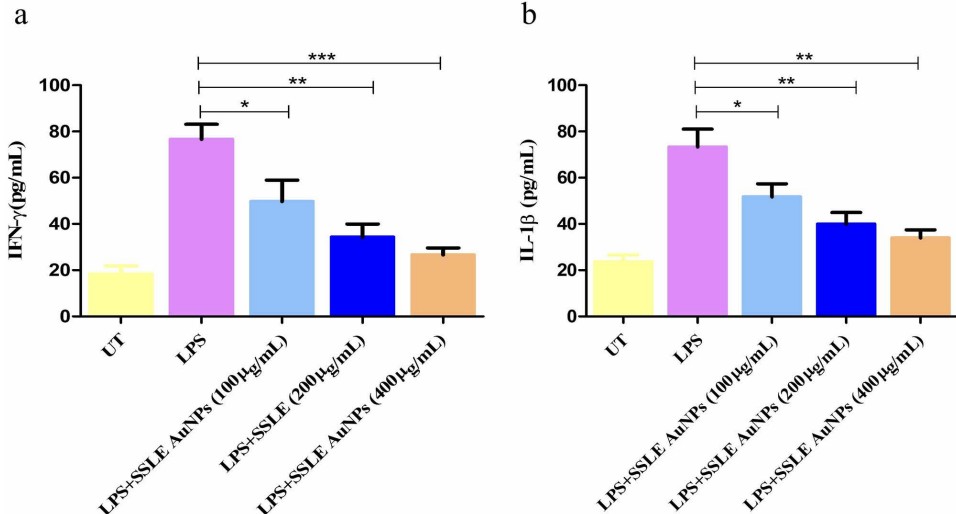

**Fig 10. Levels of pro-inflammatory cytokines (a) IFN-γ and (b) IL-1β in LPS stimulated murine alveolar macrophages J774A.1 cells post-SSLE-AuNPs exposure.** $*p < 0.05$, $**p < 0.01$ and $***p < 0.001$.

Although the reported study underlines the cross-functional attributes of SSLE-AuNPs, still certain limitations require consideration. Initially, a more focused exploration of the bioactive compounds of S. splendens involved in reduction of Au+ to Au would have been more impactful in deciphering the green synthesis potential of the plant extract. Secondly, an *in vitro* approach of the study though mechanically informative further warrants validation through *in vivo* models to further gain insights into the biodistribution, pharmacokinetics and plausible side effects of SSLE-AuNPs. Subsequently, in spite of the fact that DLS and TEM elucidated the stability and size of SSLE-AuNPs, an insight into the crystallinity of the synthesized could have provided further structural information of SSLE-AuNPs. Lastly, the diverse phytochemical composition of SSLE, although being synergistically advantageous for various bioactivities, can be relatively challenging for bioactive compound specific effects.

## 5. Conclusions

The present study elucidated the multipotent activity of AuNPs biosynthesized using *S. splendens* aqueous leaf extract (Fig 11). The results indicated potent antioxidant, antibacterial, anticancer and anti-inflammatory effects of SSLE-AuNPs in suitable *in vitro* model systems. The findings open the possibilities for further therapeutic exploration of SSLE-AuNPs, and subsequent development of nano-therapeutics specifically against lung cancer.

## Supporting information

**S1 Fig. Dried leaves, leaf powder and aqueous extract of *Salvia splendens*.**
(TIF)

## Acknowledgments

This research has been funded by Scientific Research Deanship at University of Hail-Saudi Arabia through project number (RG-23 179).

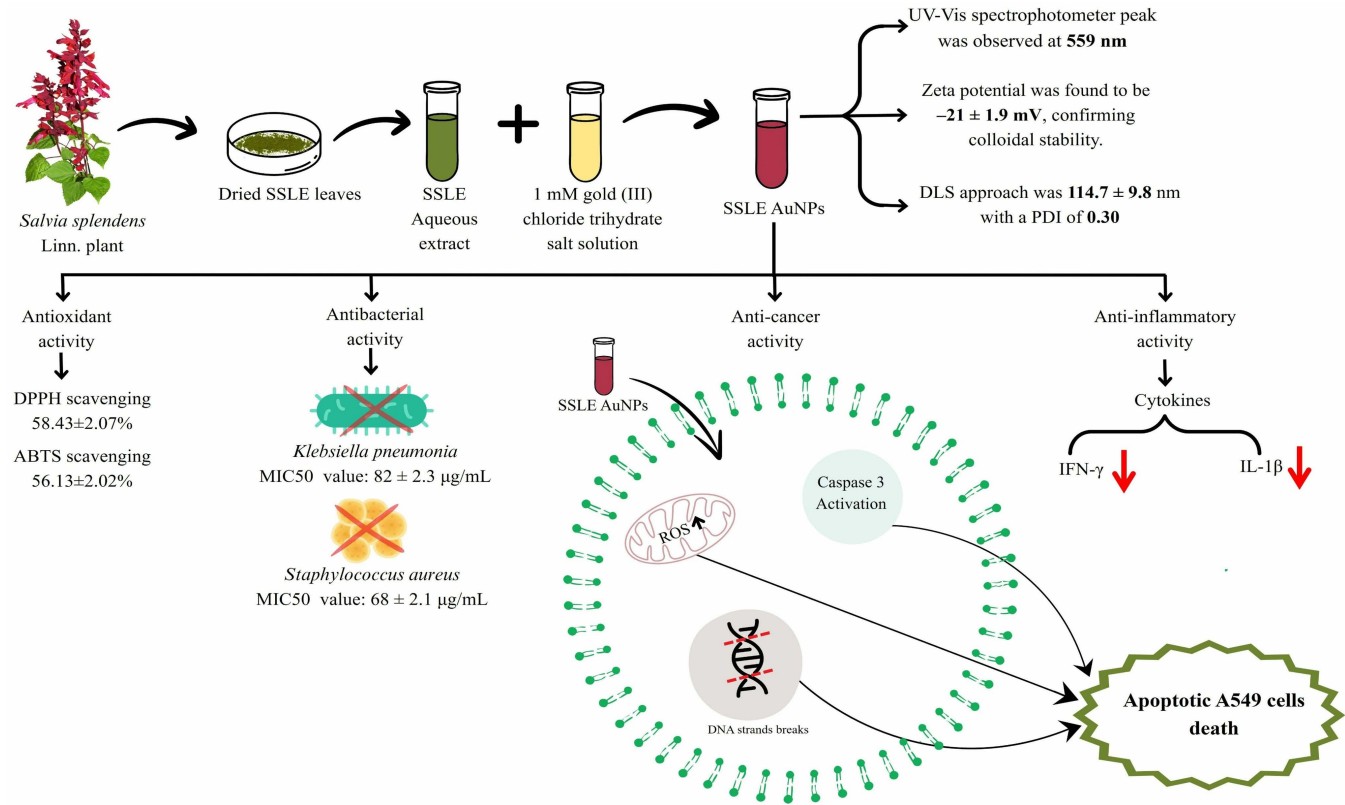

**Fig 11. Illustrative representation for synthesis of SSLE-AuNPs synthesized using *S. splendens* leaves extract.** The extract was incubated with gold (III) chloride trihydrate salt and incubated. The change in color of the extract indicated towards the formation of SSLE AuNPs. SSLE-AuNPs were further characterized through UV-Vis spectrophotometry, DLS for their relative diameter and zeta-potential followed by their investigation through TEM. SSLE AuNPs were further evaluated for their various biological attributes.

## Author contributions

**Conceptualization:** Amr Selim Abu Lila, Afrasim Moin, Syed Mohd Danish Rizvi.

**Formal analysis:** Asma Ayyed AL-Shammary, Dinesh Chandra Sharma.

**Funding acquisition:** Afrasim Moin.

**Methodology:** Nabeel Ahmad, Afza Ahmad, Syed Mohd Danish Rizvi, Rohit Kumar Tiwari.

**Project administration:** Amr Selim Abu Lila.

**Resources:** Syed Mohd Danish Rizvi, Rohit Kumar Tiwari.

**Software:** Nabeel Ahmad, Dinesh Chandra Sharma.

**Supervision:** Syed Mohd Danish Rizvi.

**Validation:** Asma Ayyed AL-Shammary, Afza Ahmad.

**Writing – original draft:** Afrasim Moin, Nabeel Ahmad, Afza Ahmad, Rohit Kumar Tiwari.

**Writing – review & editing:** Amr Selim Abu Lila, Asma Ayyed AL-Shammary, Dinesh Chandra Sharma.

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
