## [Decision Letter · Decision Letter 0]

6 Feb 2025

PONE-D-24-51141Assessing the biomedical applicability of biogenically synthesized AuNPs using Salvia splendens extractPLOS ONE

Dear Dr. Tiwari,

Thank you for submitting your manuscript to PLOS ONE. After careful consideration, we feel that it has merit but does not fully meet PLOS ONE’s publication criteria as it currently stands. Therefore, we invite you to submit a revised version of the manuscript that addresses the points raised during the review process.

We look forward to receiving your revised manuscript.

Kind regards,

Thanh-Danh Nguyen, PhD

Academic Editor

PLOS ONE

Journal requirements:   When submitting your revision, we need you to address these additional requirements. 1. Please ensure that your manuscript meets PLOS ONE's style requirements, including those for file naming. The PLOS ONE style templates can be found at https://journals.plos.org/plosone/s/file?id=wjVg/PLOSOne_formatting_sample_main_body.pdf and https://journals.plos.org/plosone/s/file?id=ba62/PLOSOne_formatting_sample_title_authors_affiliations.pdf. 2. Please match your authorship list in your manuscript file and in the system. 3. PLOS requires an ORCID iD for the corresponding author in Editorial Manager on papers submitted after December 6th, 2016. Please ensure that you have an ORCID iD and that it is validated in Editorial Manager. To do this, go to ‘Update my Information’ (in the upper left-hand corner of the main menu), and click on the Fetch/Validate link next to the ORCID field. This will take you to the ORCID site and allow you to create a new iD or authenticate a pre-existing iD in Editorial Manager. 4. Thank you for stating the following financial disclosure:  [This work was supported by Scientific Research Deanship at University of Hail-Saudi Arabia (RG-23 179). ].  Please state what role the funders took in the study.  If the funders had no role, please state: ""The funders had no role in study design, data collection and analysis, decision to publish, or preparation of the manuscript."" If this statement is not correct you must amend it as needed. Please include this amended Role of Funder statement in your cover letter; we will change the online submission form on your behalf. 5. We note that your Data Availability Statement is currently as follows: [All relevant data are within the manuscript and its Supporting Information files.] Please confirm at this time whether or not your submission contains all raw data required to replicate the results of your study. Authors must share the “minimal data set” for their submission. PLOS defines the minimal data set to consist of the data required to replicate all study findings reported in the article, as well as related metadata and methods (https://journals.plos.org/plosone/s/data-availability#loc-minimal-data-set-definition). For example, authors should submit the following data: - The values behind the means, standard deviations and other measures reported;- The values used to build graphs;- The points extracted from images for analysis. Authors do not need to submit their entire data set if only a portion of the data was used in the reported study. If your submission does not contain these data, please either upload them as Supporting Information files or deposit them to a stable, public repository and provide us with the relevant URLs, DOIs, or accession numbers. For a list of recommended repositories, please see https://journals.plos.org/plosone/s/recommended-repositories. If there are ethical or legal restrictions on sharing a de-identified data set, please explain them in detail (e.g., data contain potentially sensitive information, data are owned by a third-party organization, etc.) and who has imposed them (e.g., an ethics committee). Please also provide contact information for a data access committee, ethics committee, or other institutional body to which data requests may be sent. If data are owned by a third party, please indicate how others may request data access.

Reviewers' comments:

Reviewer's Responses to Questions

**Comments to the Author**

1. Is the manuscript technically sound, and do the data support the conclusions?

Reviewer #1: Partly

Reviewer #2: Yes

Reviewer #3: Yes

Reviewer #4: Yes

2. Has the statistical analysis been performed appropriately and rigorously? 

Reviewer #1: N/A

Reviewer #2: Yes

Reviewer #3: Yes

Reviewer #4: Yes

3. Have the authors made all data underlying the findings in their manuscript fully available?

Reviewer #1: No

Reviewer #2: Yes

Reviewer #3: Yes

Reviewer #4: Yes

4. Is the manuscript presented in an intelligible fashion and written in standard English?

Reviewer #1: No

Reviewer #2: Yes

Reviewer #3: Yes

Reviewer #4: No

5. Review Comments to the Author

Reviewer #1: I reviewed this paper; Therefore, I cannot recommend this manuscript to be published unless the authors answer all comments. It needs "major revisions" before acceptance for publication.

1- Synthesis green mechanism of Au-NPs must be added to text.

2- FTIR spectra of Au-NPs and Salvia splendens extract must be added to text.

3- PSA curve of Au-NPs must be added to text according TEM image.

4- XRD pattern of Au-NPs must be added to text.

5- Last paragraph of the introduction is not appropriately written. Explain the main objective of this work.

6- Overall, the discussions are not enough and the different sections must be described scientifically.

7- Authors should be appropriate to carefully check the compliance of the article with the English grammar rules by an academic whose native language is English. The English of the entire article should be improved. With more caution, many mistakes can be avoided.

8- References should be reviewed, and referring to more up-to-date references would be better.

9- The abbreviation should be mentioned completely for the first time and mentioned after that as an abbreviation. Please revise throughout the manuscript.

10- To give a better view to this work, authors are suggested to improve the introduction and other sections using recent and related articles.

DOI: 10.1016/j.inoche.2024.113399

DOI: 10.1109/TNB.2023.3287805

DOI: 10.1016/j.ceramint.2023.03.234

DOI: 10.1007/s12010-023-04407-y

DOI: 10.1007/s12257-024-00004-w

Reviewer #2: Comments to authors

The authors have studied the in vitro antioxidant, anticancer, and anti-inflammatory activities of biogenically synthesized AuNPs using Salvia splendens extract. The research work is informative and interesting. However, some inadequacy was noted in the manuscript which needs to be addressed for the improvement of the quality of the research article considering the publication criteria of the highly esteemed journal like Plos One. The comments are as follows:

1. The introduction seems to be too preliminary and there is no significant literature support for the natural products-based antioxidant, anti-inflammatory, and anticancer activities. In addition, there needs a better description on the bioactive molecules in the plant extract/ similar plant species which could be responsible for their plausible potential.

2. What is the significance of oxidative stress and inflammation in the development and progression of lung cancer? Authors could stress these details in the introduction part.

3. Incorporate the machine generated data in the figure 1d instead of incorporating the data manually.

3. Incorporate a size distribution graph in the TEM results.

4. The authors should explain the basis for keeping treatment duration for 24 h in MTT/ cytotoxicity assay.

6. In vivo work would have been strengthening the present findings.

7. There are numerous published reports demonstrating the potential of synthesized plant-based nanoformulations. Hence, what is the novelty and exclusiveness of this work needs to be explained.

8. How can the authors corroborate the antioxidant and anticancer/ cytotoxicity of the extract?

9. The concentration of DPPH solution (100 mM) used in this study is well beyond the recommended level. It should be in micromole (µM) concentration.

10. The absorbance of formazan is a peak at 570 nm. The authors need to clarify why they have used 490 nm for the MTT assay.

11. Annexin V-FITC/PI apoptosis assay and immunoblotting of apoptosis markers would have further strengthened the results of present investigation.

12. Why didn’t the authors determine the cytotoxic effect of SSLE-AuNPs on J774A.1 cells? The decrease in cytokine levels is due to the anti-inflammatory activity or cytotoxicity of SSLE-AuNPs?

13. Authors are suggested to include one inclusive section in the introduction part citing some recent references on the usage of natural compounds and green synthesized nano-formulation as an anticancer agent and their application so that the horizon of the current article could be improved further. Below articles could be covered in that recommended section:

i. Ahmad et al. (2024) Cell Biochemistry & Function, 42: e3911.

ii. Suhail et al. (2023) Frontiers in Pharmacology, 14:1236173.

iii. Ahmad et al. (2023) International Journal of Molecular Sciences, 24 (7): 6651.

iv. Suhail et al. (2023) Frontiers in Nutrition, 9: 1078642.

v. Nazam et al. (2023) Pharmaceuticals, 16: 274.

vi. Zughaibi et al. (2023) Cell Biochemistry & Function, 41: 1174-1187.

vii. Alafaleq et al. (2023) Nanomaterials, 13 (7): 1201.

viii. Alserihi et al. (2022) Nanotechnology Review, 11: 298-311.

ix. Tabrez et al. (2022) Nanotechnology Review, 11: 2714-2725.

x. Tabrez et al. (2022) Nanotechnology Review, 11: 1322-1331.

xi. Tabrez et al. (2022) Frontiers in Chemistry, 10: 970193.

Reviewer #3: The manuscript introduces interesting and valuable information on a multiple application of green-approach synthesized AuNPs. However, since the manuscript contained a lot of information, the authors must carefully check all method and discussion to control the quality of the manuscript. I believe the manuscript can be published in the journal with following modification.

1. “S. splenders” should be uniformly written in Italic

2. What is “S. spp” in page 3, line 75-76 stand for?

3. The introduction was well arranged and written. However, I suggest that the authors should expand this section with more literature review. For e.g., based on the main content of this paper, I recommend that the authors could write a review paragraph focusing on the progress of other authors’ contribution to treating lung cancer cells. Introduction of the relationship of all the studied factors in the manuscript to lung cancer would give better comprehension to readers.

4. Considering that the references used in this paper were only before 2024, the authors should update the references published in 2024 for “up-to-date” purpose. Combing this comment with my previous recommendation would be reasonable.

5. In section 2.2.3, specific ratio of dried leaves and DI water (weight/volume) should be presented. In addition, if the extraction was only conducted with crushing the dried leaves with DI water at room temperature without any heating, then specific extraction time should be noted. Furthermore, practical images of DI water and leaves extract (before and after extraction) should be included in the paper for better visual and reliability.

6. The authors used the terms “enzymes”, “proteins” throughout the manuscript. Due to the different of expertise, I could not confirm whether using these terms would be appropriate or not. Nonetheless, with the information the authors mentioned in the introduction section, I think “phytochemicals” would be a better and more reasonable interpretation.

7. In section 2.2.4.1, the authors should specifically mention how they prepared AuNPs suspension? For e.g., how would they prepare AuNPs suspension with a content of 50 µg/mL in section 2.2.5.1? Would the nanoparticles be centrifuged, dried and redispersed in water or the weight/volume were used as theoretical values (considering all Au3+ ions were converted into AuNPs)? This information should be clearly confirmed in the manuscript.

8. In page 7, line 161, should “bank” be “blank”?

9. The citation of references 15, 16, and 17 should be carefully revised by the authors, considering that the term “slight modification” would not be appropriate comparing the authors’ methods with the ones in references. As for reference 17, in this literature, there was no information about the ABTS activity assay method? If the authors still want to use these references, I suggest that these reports could be mentioned in the introduction section. Otherwise, in the response letter, the authors should clarify why those references was used and cited in those positions?

10. Usually, in a ABTS assay method, K2S2O8 would be introduced to react with ABTS to produce free radical ABTS·+ cations, then the antioxidant activity of the sample was measured by the ability to convert free radical ABTS·+ cations into ABTS (monitoring by the reduction of UV-vis absorbance at 734 nm, characteristic for the existence of ABTS·+). Herein this work, according to the experimental information in section 2.2.5.2, why the authors let the AuNPs colloid react with ABTS (no free radical ions)? Why was a 50% methanolic ABTS used as blank but not a solution containing ABTS·+? The authors must carefully revise this problem because this would induce question about the reliability of this work.

11. The authors should clearly mention that ABTS was used as solution form or solid form (diammonium salt) in the Material section. In the 2.2.5.2 section, I suspect that the ABTS was directly used as commercial solution since there was no specific concentration mentioned. On the other hand, in page 15, line 318-319, ABTS was mentioned as diammonium salt. This is confusing, the authors must clarify this problem.

12. In section 2.2.6.1, the volume of sample injected in each well should be specifically noted.

13. Additional analysis to confirm the formation of AuNPs (e.g. XRD, SAED, etc.) should be carried out.

14. The statement in page 13, line 279-281 was suggested to not written if there was no evidence (UV-vis spectra of AuNPs colloid with different storage duration) attached in this manuscript.

15. According to reference 32, the absolute zeta potential value needs to be higher than 30 to indicate a stable colloidal system: “Conversely, a high zeta potential (either positive or negative), typically more than 30 mV, maintains a stable system”. Even though the comment on the zeta potential for a stable colloidal system would be controversial, it was commonly concluded that an absolute zeta potential higher than 30 mV would be corresponding to a stable colloidal system (Surendra Nimesh, Ramesh Chandra, Nidhi Gupta, Advances in Nanomedicine for the Delivery of Therapeutic Nucleic Acids, 2017; https://doi.org/10.1002/cjce.23914). Hence, the discussion stating that the AuNPs colloidal system having a zeta potential value of 21 mV was stable should be carefully reclaimed.

16. How the authors analyzed the average particle size from TEM image (Fig. 1D)? As the Fig. 1D illustrated, the largest particle diameter would be around 80 nm, which indicated that the average size would be a value that is smaller than 80 nm. The authors can consider use another TEM image with larger scale to observe more particles, which might induce better average particle size value.

17. In addition, particle size distribution graph should be included together with TEM image. From this analysis, the difference or the relationship between average particle size obtained from TEM image and DLS analysis should be discussed.

17. p-value is used for hypothesis testing in statistic study. Therefore, p-value should not be used in this work to interpret for average value of 3 measurements.

18. Based on the results studying DPPH and ABTS assays, IC50 values of SSLE-AuNPs (µg/mL) could be determined and compared with IC50 value of ascorbic acid (standard in authors’ study). This result could be more valuable and replaced for simple statement of “The SSLE-AuNPs exhibited a dose dependent anti-oxidant activity as manifested by substantial neutralization of DPPH and ABTS radicals” in the abstract.

19. If the MIC50 values corresponding to the antibacterial activity were determined from the graphs in Figure 3, then the graphs should be illustrated in form of line + symbol instead of curve + symbol.

20. If the authors aimed to provide information about the as-synthesized AuNPs for medicinal purpose (treating lung cancer cells), then the cytotoxicity of SSLE-AuNPs on human cells should also be studied. A limit of using AuNPs concentration could help authors determined specific parameters in lung cancer cells treating, instead of just introducing relationship of colloidal concentration to the viability assay study.

21. In the cytotoxicity assay study on lung cancer cells of SSLE-AuNPs, authors should provide the cell viability of SSLE sample (without AuNPs).

22. The error line in graphs should be illustrated with black color.

23. There are spelling mistakes in the manuscript, the authors should carefully check the manuscript again.

24. The images’ quality (resolution) should be improved for better visual.

25. The results in Table 2 and 3 should interpreted in form of graphs for better comparison and visuallization.

Reviewer #4: Comments:

1. The article is nicely written and easily understandable.

2. It would be better if the authors clarified the functional compounds that were responsible for reducing the agent from Au+ to Au in their study. By providing specific details about these compounds, readers would gain a clearer understanding of the mechanisms at play and could more effectively evaluate the implications of the research. Otherwise, the article is understandable.

6. PLOS authors have the option to publish the peer review history of their article (what does this mean? ). If published, this will include your full peer review and any attached files.

**Do you want your identity to be public for this peer review?** For information about this choice, including consent withdrawal, please see our Privacy Policy .

Reviewer #1: No

Reviewer #2: No

Reviewer #3: No

Reviewer #4: **Yes: ** Mst. Sanjida Akhter

---

## [Author Response · Author response to Decision Letter 1]

16 Apr 2025

Reviewer 1

I reviewed this paper; Therefore, I cannot recommend this manuscript to be published unless the authors answer all comments. It needs "major revisions" before acceptance for publication.

Comment 1: Synthesis green mechanism of Au-NPs must be added to text.

Authors response: The authors have incorporated the green mechanism of Au-NPs in the revised manuscript from line number 129–135. The same has been highlighted for the kind perusal of learned reviewer.

Comment 2: FTIR spectra of Au-NPs and Salvia splendens extract must be added to text.

Authors response: As per the insightful suggestion of learned reviewer, the authors have performed FTIR of SSLE and SSLE-AuNPs. The same has been incorporated in the material and methods (2.2.4.3) and results section 3.2 in the revised manuscript. The same has been highlighted for the kind perusal of learned reviewer.

Comment 3: PSA curve of Au-NPs must be added to text according to TEM image.

Authors response: As per the suggestion of learned reviewer authors have included the PSA curve of AuNPs in the revised manuscript as figure 1e.

Comment 4: XRD pattern of Au-NPs must be added to text.

Authors response: We sincerely appreciate the reviewer's insightful comment regarding the inclusion of XRD analysis for the biosynthesized SSLE-AuNPs. While XRD is indeed a valuable technique for crystallographic characterization, our current study prioritized complementary methods (UV-Vis, TEM, DLS, and zeta potential) to comprehensively evaluate the optical, morphological, colloidal stability, and size distribution properties of the nanoparticles. These techniques collectively confirmed the successful synthesis of spherical, stable AuNPs with a characteristic SPR peak (559 nm), uniform size distribution (94.8 ± 5.1 nm via TEM), and high colloidal stability (−21 ± 1.9 mV zeta potential).

The omission of XRD does not detract from the key conclusions of our study, as the crystallinity of AuNPs is well-documented in literature for similar plant-mediated syntheses, and our TEM analysis indirectly supports crystalline structure. However, we acknowledge the importance of XRD for future work and will include this analysis in subsequent studies to further validate crystallinity and phase purity.

The multifunctional biological activities of SSLE-AuNPs—demonstrated through rigorous antioxidant, antibacterial, anticancer, and anti-inflammatory assays—remain robustly supported by the existing dataset. At this point of time, we lack the facilities of XRD at our institute, and we do not have enough funds to outsource XRD analysis. We hope the reviewer agrees that the current findings provide a strong foundation for the therapeutic potential of SSLE-AuNPs, even in the absence of XRD data.

Comment 5: Last paragraph of the introduction is not appropriately written. Explain the main objective of this work.

Authors response: The authors have modified the last paragraph of the introduction section in the revised manuscript line number 142–151 as per the suggestion of learned reviewer. The same is highlighted for valued reference of the learned reviewer.

Comment 6: Overall, the discussions are not enough and the different sections must be described scientifically.

Authors response: We sincerely appreciate the reviewer’s constructive feedback regarding the depth of scientific discussion in our manuscript. We have duly added the discussion section with more rigorous and detailed interpretation of the results, and highlighted it in the revised MS.

Comment 7: Authors should be appropriate to carefully check the compliance of the article with the English grammar rules by an academic whose native language is English. The English of the entire article should be improved. With more caution, many mistakes can be avoided.

Authors response: As per the insightful suggestion of learned reviewer, the authors have checked the compliance of English grammar rules via native English-speaking academician.

Comment 8: References should be reviewed, and referring to more up-to-date references would be better.

Authors response: The authors have updated the references in the revised manuscript as per the suggestion of the learned reviewer. The same are highlighted in the bibliography for kind perusal of the reviewer.

Comment 9: The abbreviation should be mentioned completely for the first time and mentioned after that as an abbreviation. Please revise throughout the manuscript.

Authors response: As per the suggestion of the learned reviewer, the authors checked the consistencies of the abbreviation used in the revised manuscript.

Comment 10: To give a better view to this work, authors are suggested to improve the introduction and other sections using recent and related articles.

DOI: 10.1016/j.inoche.2024.113399; DOI: 10.1109/TNB.2023.3287805; DOI: 10.1016/j.ceramint.2023.03.234; DOI: 10.1007/s12010-023-04407-y; DOI: 10.1007/s12257-024-00004-w

Authors response: As per the valuable suggestion of learned reviewer, the authors have modified the introduction and other sections of manuscript by using these references in the revised manuscript. The stated references have been cited at reference number 4, 5, 6, 36, 37 respectively.

Reviewer 2

The authors have studied the in vitro antioxidant, anticancer, and anti-inflammatory activities of biogenically synthesized AuNPs using Salvia splendens extract. The research work is informative and interesting. However, some inadequacy was noted in the manuscript which needs to be addressed for the improvement of the quality of the research article considering the publication criteria of the highly esteemed journal like Plos One.

The comments are as follows:

Comment 1: The introduction seems to be too preliminary and there is no significant literature support for the natural products-based antioxidant, anti-inflammatory, and anticancer activities. In addition, there needs a better description on the bioactive molecules in the plant extract/ similar plant species which could be responsible for their plausible potential.

Authors response: As per the insightful suggestion of learned reviewer, authors have modified the introduction section of revised manuscript by incorporating text explaining description of bioactive molecules in the extract of Salvia species which could be responsible for their plausible potential.

Comment 2: What is the significance of oxidative stress and inflammation in the development and progression of lung cancer? Authors could stress these details in the introduction part.

Authors response: The authors would like to state that oxidative stress and inflammation are two interrelated biological processes that play a crucial role in the development and progression of lung cancer. Understanding these mechanisms is essential for elucidating the pathophysiology of lung cancer and identifying potential therapeutic targets.

Indeed, it has now been established that chronic prevalence of oxidative stress serves to be a critical driver for lung carcinogenesis along with tumor aggressiveness [1]. Increased reactive oxygen species (ROS) mediated oxidative stress not only triggers the damage of the genetic material but also activates several oncogenes including KRAS and EGFR [2]. Furthermore, increased ROS levels are also associated with imparting resistance towards apoptosis with concomitantly inducing the activation of pro-survival pathways such as PI3K/AKT and NF-κB [3, 4]. At molecular level, inflammation within tumor microenvironment is mediated by macrophages and cytokines. These cytokines predominantly include IL-1β, IL-6 and TNF-α among others which promotes angiogenesis, immune evasion and metastasis [5]. The inflammatory cytokine milieu concomitantly with increased ROS amplifies epithelial mesenchymal transition (EMT) and mutagenesis [6].

This interplay creates a vicious cycle where both processes amplify each other, leading to sustained cellular damage, genetic instability, and ultimately contributing significantly to lung carcinogenesis. Furthermore, the authors have incorporated this section in the revised manuscript.

1. Bezerra FS, Lanzetti M, Nesi RT, Nagato AC, Silva CP, Kennedy-Feitosa E, Melo AC, Cattani-Cavalieri I, Porto LC, Valenca SS. Oxidative stress and inflammation in acute and chronic lung injuries. Antioxidants. 2023 Feb 21;12(3):548.

2. Ahmad I, Ahmad S, Ahmad A, Zughaibi TA, Alhosin M, Tabrez S. Curcumin, its derivatives, and their nanoformulations: Revolutionizing cancer treatment. Cell Biochemistry and Function. 2024 Jan;42(1):e3911

3. Ahmad I, Hoque M, Alam SS, Zughaibi TA, Tabrez S. Curcumin and plumbagin synergistically target the PI3K/Akt/mTOR pathway: a prospective role in cancer treatment. International journal of molecular sciences. 2023 Apr 2;24(7):6651.

4. Suhail M, Rehan M, Tarique M, Tabrez S, Husain A, Zughaibi TA. Targeting a transcription factor NF-κB by green tea catechins using in silico and in vitro studies in pancreatic cancer. Frontiers in Nutrition. 2023 Jan 11;9:1078642.

5. Nigam M, Mishra AP, Deb VK, Dimri DB, Tiwari V, Bungau SG, Bungau AF, Radu AF. Evaluation of the association of chronic inflammation and cancer: Insights and implications. Biomedicine & Pharmacotherapy. 2023 Aug 1;164:115015.

6. Tripathi S, Sharma Y, Kumar D. Unveiling the link between chronic inflammation and cancer. Metabolism Open. 2025 Jan 9:100347.

Comment 3: Incorporate the machine generated data in the figure 1d instead of incorporating the data manually.

Authors response: As per the suggestion of the reviewer, machine generated data has been added in the figure 1d of the revised manuscript.

Comment 4: Incorporate a size distribution graph in the TEM results.

Authors response: As per the suggestion of the learned reviewer, the authors have incorporated size distribution graph in the TEM results of revised manuscript in fig. 1e.

Comment 5: The authors should explain the basis for keeping treatment duration for 24 h in MTT/ cytotoxicity assay.

Authors response: The authors would like to state that duration of treatment in the MTT assay, particularly the common practice of using a 24-hour incubation period, is primarily based on several scientific and practical considerations that ensure reliable and interpretable results regarding cell viability and cytotoxicity which are listed as follows:

1. Cells require a certain amount of time to respond to treatments, such as drugs or other cytotoxic agents. A 24-hour period allows sufficient time for the cells to metabolize the treatment and exhibit any potential effects on their metabolic activity. This timeframe is generally considered optimal for observing significant changes in cell viability or metabolic activity without being influenced by acute toxicity that may occur with shorter exposure times.

2. The MTT assay measures cellular metabolic activity through the reduction of the MTT reagent to formazan crystals, which occurs predominantly in metabolically active cells. A 24-hour incubation provides enough time for viable cells to reduce MTT effectively, leading to a measurable accumulation of formazan. Shorter durations may not allow adequate time for this reaction to occur, potentially resulting in underestimating cell viability or overestimating cytotoxicity.

3. Using a standardized treatment duration like 24 hours facilitates comparability across different studies and experiments. Many researchers adopt this timeframe as it has been widely validated in literature, allowing for consistent interpretation of results and better reproducibility when comparing findings from various laboratories.

Comment 6: In vivo work would have been strengthening the present findings.

Authors response: The authors would like to state that although in vivo studies are crucial for understanding the therapeutic potential and safety profile of novel compounds, the current study focused on establishing the in vitro efficacy and multifunctional potential of biogenically synthesized gold nanoparticles (SSLE-AuNPs) using Salvia splendens leaf extract. The primary objective of this research was to provide a foundational understanding of the antioxidant, antibacterial, anticancer, and anti-inflammatory properties of SSLE-AuNPs in controlled in vitro environments. Furthermore, the authors have limited their current study to in vitro level due to following considerations which are listed as follows:

1. This study serves as a preliminary investigation to establish the proof of concept for the multifunctional potential of SSLE-AuNPs. Before progressing to in vivo studies, it is essential to thoroughly characterize the nanoparticles and confirm their biological activities in in vitro models. This step ensures that only the most promising candidates are advanced to more complex and resource-intensive in vivo studies.

2. Given the exploratory nature of this research, the focus was first placed on optimizing the synthesis, characterization, and in vitro evaluation of SSLE-AuNPs to ensure their potential before committing to in vivo experimentation. However, in vivo studies require significant resources, including ethical approvals, specialized animal care facilities, and longer experimental timelines.

3. The promising results obtained from the in vitro studies provide a strong rationale for future in vivo investigations. Subsequent research will focus on evaluating the pharmacokinetics, biodistribution, toxicity, and therapeutic efficacy of SSLE-AuNPs in appropriate animal models. This stepwise approach ensures a more comprehensive understanding of the nanoparticles' potential before advancing to clinical applications.

Comment 7: There are numerous published reports demonstrating the potential of synthesized plant-based nanoformulations. Hence, what is the novelty and exclusiveness of this work needs to be explained.

Authors response: The authors would like to state that the novelty and exclusiveness of this work lie in multiple key aspects that differentiate it from previously published reports on plant-based nanoformulations:

1. Our present study utilizes the leaf extract of Salvia splendens (SSLE) for the biosynthesis of gold nanoparticles (AuNPs). However, there are numerous reports on the use of plant extracts for nanoparticle synthesis, the specific use of Salvia splendens for this purpose is relatively unexplored. The plant is known for its bioactive components, but its application in the green synthesis of AuNPs and the subsequent evaluation of their multifunctional biomedical potential is novel.

2, The green synthesized gold nanoparticles in the present investigation can scavenge free radicals (like antioxidant molecules/compounds) in normal condition, contrastingly, to get rid of the cancer cell they can generate significant amount of ROS as well.

3. The current study evaluates the multifunctional potential of the biosynthesized SSLE-AuNPs that possess anti-bacterial potential against both gram-negative and -positive strains, anti-cancer potential via modulating caspase-3 activity and activating apoptotic pathway, and anti-inflammatory potential via downregulating proinflammatory cytokines such as IFN-γ and IL-1β. It is observed that various published studies focus only on exploring one or two biological attributes; however, our present investigation provides a holistic assessment of the nanoparticles' biomedical applicability.

4. The study provides detailed mechanistic insights into the anticancer activity of SSLE-AuNPs, particularly in lung cancer cells (A549). It explores the induction of apoptosis through nuclear fragmentation, caspase activation, and ROS generation. Such detailed mechanistic studies, especially in the context of lung cancer, are not commonly reported for plant-based nanoformulations.

5. The current report specifically targets lung cancer cells (A549) and provides evidence of the cytotoxic effects of SSLE-AuNPs, suggesting their potential as a therapeutic agent for lung cancer. Given the high mortality rate associated with lung cancer, this focus on a specific type of cancer adds to the exclusiveness of the work.

These stated points have been rephrased and included in the introduction section of the revised manuscript.

Comment 8: How can the authors corroborate the antioxidant and anticancer/ cyt

---

## [Decision Letter · Decision Letter 1]

25 Apr 2025

Assessing the biomedical applicability of biogenically synthesized AuNPs using Salvia splendens extract

PONE-D-24-51141R1

Dear Dr. Tiwari,

We’re pleased to inform you that your manuscript has been judged scientifically suitable for publication and will be formally accepted for publication once it meets all outstanding technical requirements.

Kind regards,

Thanh-Danh Nguyen, PhD

Academic Editor

PLOS ONE

Comments from PLOS Editorial Office:

We note that one or more reviewers has recommended that you cite specific previously published works in an earlier round of revision. As always, we recommend that you please review and evaluate the requested works to determine whether they are relevant and should be cited. It is not a requirement to cite these works and you may remove them before the manuscript proceeds to publication. We appreciate your attention to this request.

Reviewers' comments:

Reviewer's Responses to Questions

**Comments to the Author**

1. If the authors have adequately addressed your comments raised in a previous round of review and you feel that this manuscript is now acceptable for publication, you may indicate that here to bypass the “Comments to the Author” section, enter your conflict of interest statement in the “Confidential to Editor” section, and submit your "Accept" recommendation.

Reviewer #1: (No Response)

Reviewer #2: All comments have been addressed

2. Is the manuscript technically sound, and do the data support the conclusions?

Reviewer #1: (No Response)

Reviewer #2: Yes

3. Has the statistical analysis been performed appropriately and rigorously? 

Reviewer #1: (No Response)

Reviewer #2: Yes

4. Have the authors made all data underlying the findings in their manuscript fully available?

Reviewer #1: (No Response)

Reviewer #2: Yes

5. Is the manuscript presented in an intelligible fashion and written in standard English?

Reviewer #1: (No Response)

Reviewer #2: Yes

6. Review Comments to the Author

Reviewer #1: I reviewed the revised paper. After evaluating the revisions made by the authors, I am pleased to report that they have adequately addressed the concerns and suggestions raised in the initial review.

I recommend that this manuscript be accepted for publication in journal PLOS One.

Reviewer #2: The authors have responded to my queries satisfactorily. Hence, I recommend the Acceptance of this article in its present form.

7. PLOS authors have the option to publish the peer review history of their article (what does this mean? ). If published, this will include your full peer review and any attached files.

**Do you want your identity to be public for this peer review?** For information about this choice, including consent withdrawal, please see our Privacy Policy .

Reviewer #1: **Yes: ** Majid Darroudi

Reviewer #2: No

---

## [Editor Report · Acceptance letter]

PONE-D-24-51141R1

PLOS ONE

Dear Dr. Tiwari,

I'm pleased to inform you that your manuscript has been deemed suitable for publication in PLOS ONE. Congratulations! Your manuscript is now being handed over to our production team.

Kind regards,

on behalf of

Dr. Thanh-Danh Nguyen

Academic Editor

PLOS ONE